# Error analyses of a multistatic meteor radar system to obtain a 3-dimensional spatial resolution distribution

Wei Zhong[1,2], Xianghui Xue[1,2,3], Wen Yi[1,2], Iain M. Reid[4,5] Tingdi Chen[1,2,3], Xiankang Dou[1,6]

[1]CAS Key Laboratory of Geospace Environment, Department of Geophysics and Planetary Sciences, University of Science and Technology of China, Hefei, China
[2]Mengcheng National Geophysical Observatory, School of Earth and Space Sciences, University of Science and Technology of China, Hefei, China
[3]CAS Center for Excellence in Comparative Planetology, Hefei, China
[4]ATRAD Pty Ltd., Thebarton, South Australia, Australia
[5]School of Physical Sciences, University of Adelaide, Adelaide, South Australia, Australia
[6]Wuhan University, Wuhan, China

*Correspondence to*: Xianghui Xue (xuexh@ustc.edu.cn)

**Abstract**:In recent years, the concept of multistatic meteor radar systems has attracted the attention of the atmospheric radar community, focusing on the mesosphere and lower thermosphere (MLT) region. Recently, there have been some notable experiments using such multistatic meteor radar systems. Good spatial resolution is vital for meteor radars because nearly all parameter inversion processes rely on the accurate location of the meteor trail specular point. It is timely then for a careful discussion focussed on the error distribution of multistatic meteor radar systems. In this study, we discuss the measurement errors that affect the spatial resolution and obtain the spatial resolution distribution in 3-dimensional space for the first time. The spatial resolution distribution can both help design a multistatic meteor radar system and improve the performance of existing radar systems. Moreover, the spatial resolution distribution allows the accuracy of retrieved parameters such as the wind field to be determined.

## 1 Introduction

The mesosphere and lower thermosphere (MLT) is a transition region from the neutral to the partially ionized atmosphere. It is dominated by the effects of atmospheric waves, including planetary waves, tides and gravity waves. It is also a relatively poorly sampled part of the Earth's atmosphere by ground-based instruments. One widely used approach to sample this region is the meteor radar technique. The ablation of incoming meteors in the MLT region, i.e., ~80 – 110 km, creates layers of metal atoms, which can be observed from the ground by photometry or lidar (Jia et al., 2016; Xue et al., 2013). During meteor ablation, the trails caused by small meteor particles provide a strong atmospheric tracer within the MLT region that can be continuously detected by meteor radars, regardless of weather conditions. Consequently, the meteor radar technique has been a powerful tool for studying the MLT region for decades (Hocking et al., 2001; Holdsworth et al., 2004; Jacobi et al., 2008; Stober et al., 2013; Yi et al., 2018). Most modern meteor radars are monostatic, and this has two main limitations in retrieving

the complete wind fields. Firstly, limited meteor rates and relatively low measurement accuracies necessitate that all measurements in the same height range are processed to calculate a "mean" wind. Secondly, classic monostatic radars retrieve winds based on the assumption of a homogenous wind in the horizontal and usually a zero wind in the vertical direction.

The latter conditions can be partly relaxed if the count rates are high and the detections are distributed through a representative range of azimuths. If this is the case, a version of a Velocity Azimuth Display (VAD) analysis can be applied by expanding the zonal and meridional winds using a truncated Taylor expansion (Browning and Wexler, 1968). This is because each valid meteor detection yields a radial velocity in a particular viewing direction of the radar. The radar is effectively a multi-beam Doppler radar where the "beams" are determined by the meteor detections. If there are enough suitably distributed detections

in azimuth in a given observing period, the Taylor expansion approach using cartesian coordinates yields the mean zonal and meridional wind components ($u_0, v_0$), the horizontal divergence $\left(\frac{\partial u}{\partial x} + \frac{\partial v}{\partial y}\right)$, the stretching $\left(\frac{\partial u}{\partial x} - \frac{\partial v}{\partial y}\right)$ and the shearing $\left(\frac{\partial u}{\partial y} + \frac{\partial v}{\partial x}\right)$ deformations of the wind fields from an analysis of the radial velocities. However, because the radar can only retrieve the wind projection in the radial direction as measured from the radar, the vorticity $\left(\frac{\partial v}{\partial x} - \frac{\partial u}{\partial y}\right)$ of the wind fields is not available. This is common to all monostatic radar systems and a discussion of measurable parameters in the context of

multiple fixed beam upper atmosphere Doppler radars is given by (Reid, 1987). Even by relaxing the assumption of a homogeneous wind fields and using the more advanced Volume Velocity Processing (VVP) (Philippe and Corbin, 1979) to retrieve the wind fields, the horizontal gradients of the wind fields cannot be recovered due to the lack of vorticity information. To obtain a better understanding of the spatial variation of the MLT region wind fields, larger area observations (and hence higher meteor count rates) and sampling the observed area from different viewing angles are needed. An extension of the

classic monostatic meteor technique is required to satisfy these needs.

To resolve the limitations outlined above, the concept of multistatic meteor radar systems, such as MMARIA (multi-static and multi-frequency agile radar for investigations of the atmosphere) (Stober and Chau, 2015) and SIMO (single input multiple output) (Spargo et al., 2019), MIMO (multiple input multiple output radar) (Chau et al., 2019; DOREY et al., 1984) have been designed and implemented (Stober et al., 2018). Multistatic systems can utilize the forward scatter of meteor trails, thus

providing another perspective for observing the MLT. Multistatic meteor radar systems have several advantages over classic monostatic meteor radars, such as obtaining higher-order wind field information and covering wider observation areas. There have been some particularly innovative studies using multistatic meteor radar systems in recent years. For example, by combining MMARIA and the continuous wave multistatic radar technique (Vierinen et al., 2016), Stober et al. (2018) built a 5-station 7-link multistatic radar network covering an approximately 600 km×600 km observing region over Germany to

retrieve an arbitrary non-homogenous wind field with a 30 km×30 km horizontal resolution. Chau et al. (2017) used two adjacent classic monostatic specular meteor radars in northern Norway to obtain horizontal divergence and vorticity. Other approaches, such as coded continuous wave meteor radar (Vierinen et al., 2019) and the compressed sense method in MIMO sparse signal recovery (Urco et al., 2019) are described in the corresponding references.

Analysing spatial resolution limits is a fundamental but difficult topic for meteor radar systems. Meteor radar systems transmit and then receive radio waves reflected from meteor trails using a cluster of receiving antennas; commonly five antennas as in the Jones et al. (1998) configuration. By analysing the cross correlations of the signals received on several pairs of antennas, the angle of arrival (AoA) of each return can be determined. The AoA is described by the zenith angle $\theta$ and azimuth angle $\phi$. By measuring the wave propagation time from the meteor trail, range information can be determined. Most meteor radar systems rely on specular reflections from meteor trails. Thus, by combining the AoA and the range information and then using geometric analysis the location of a meteor trail can be determined. Accurately locating the meteor trail specular point (MTSP hereinafter) is important since atmospheric parameter retrieval (such as the wind field or the temperature) depends on the location information of meteor trails. The location accuracy, namely the spatial resolution, determines the reliability of the retrieved parameters. For multistatic meteor radar systems which can relax the assumption of a homogenous horizontal wind field, the location accuracy becomes a more important issue because the horizontal spatial resolution affects the accuracy of the retrieved horizontal wind field.

Although meteor radar systems have developed well experimentally in recent years, the reliability of the retrieved atmospheric parameters still requires further investigation for both the monostatic and multistatic meteor radar cases. In an attempt to investigate errors in two radar techniques, Wilhelm et al. (2017) compared 11 years of MLT region wind data from a partial reflection (PR) radar with collocated monostatic meteor radar winds and determined the 'correction factors' to bring the winds into agreement. Reid et al. (2018) reported a similar study for two locations for data obtained over several years. While the comparisons are interesting, partial reflection radars operating in the medium frequency (MF) and lower high frequency (HF) bands produce a height dependent bias in the measured winds (see e.g., Reid, 2015) which limits the ability to estimate errors in the meteor winds by comparing with them. However, the PR radar technique is one of very few that provides day and night coverage and data rates in the MLT comparable to that of meteor radars.

Meteor radars have largely replaced PR radars for MLT studies and are generally regarded as providing reference quality winds. It is essential then to know the reliability of atmospheric parameters determined by meteor radars and to do this, some quantitative error analyses are necessary.

A number of recent studies have discussed AoA measurement errors for meteor radars (Kang, 2008; Vaudrin et al., 2018; Younger and Reid, 2017). These studies focus on the phase errors in receiver antenna pairs; Younger and Reid for the monostatic case, and Vaudrin et al. for a more general case which included multistatic meteor radars. Hocking (2018) used another approach and developed a vertical resolution analysis method for the 2-dimensional bistatic case. Hocking's method (HM hereinafter) simplifies the error propagation process in the receiving antennas and puts emphasis on how a bistatic meteor radar configuration affects the vertical resolution in a vertical section. It does not consider the radial distance measuring error. In this paper, we consider the more general 3-dimensional case and determine the spatial distribution of both the horizontal and vertical resolution uncertainties.

We analyse the multistatic meteor radar resolution distribution in a three-dimensional space for both vertical and horizontal resolution for the first time. This spatial resolution is a prerequisite for evaluating the reliability of retrieved atmospheric parameters, such as the wind field and the temperature.

## 2 Analytical Method

### 2.1 Brief introduction

The HM will be introduced briefly here to help understand our generalization. In the HM, measurement errors that affect the vertical resolution can be classified into two types: one caused by the zenith angle measuring error $\delta\theta$ and one caused by the pulse-length effect on the vertical resolution. The receiving array is a simple antenna pair that is collinear with the baseline (figure 1). The HM calculates the vertical resolution in a two-dimensional vertical section which passes through the baseline. The receiver antenna pair is equivalent to one receiver arm in a Jones configuration which is comprised of three collinear antennas usually in a $2\lambda\backslash2.5\lambda$ spacing. The phase difference of the received radio wave between the receiving antenna pair is denoted as $\Delta\Psi$. In meteor radar systems, there is generally an 'acceptable' phase difference measuring error (PDME hereinafter) $\delta(\Delta\Psi)$. A higher value of $\delta(\Delta\Psi)$ means that more detected signals will be judged as meteor events, but with more misidentifications and bigger errors as well. $\delta(\Delta\Psi)$ is set to approximately $30°$ (Hocking, 2018; Younger and Reid, 2017) in most meteor radar systems. In the HM, the zenith angle measuring error $\delta\theta$ is due to $\delta(\Delta\Psi)$ and $\delta(\Delta\Psi)$ is a constant. Therefore, the error propagation in the receiver is very simple, and $\delta\theta$ is inversely proportional to the cosine of the zenith angle.

We now introduce our analytical method. Our method considers a multistatic system with multiple transmitters and one receiving array in 3-dimensional space as shown in figure 2. The receiving array is in the Jones configuration, that is, "cross-shaped", but it may also be "T-shaped" or "L-shaped". The five receiver antennas are in the same horizontal plane and constitute two orthogonal antenna arms. To avoid a complex error propagation process in the receiving array and to place emphasis on multistatic configurations, the PDMEs in the two orthogonal antenna arms ($\delta(\Delta\Psi_1)$ and $\delta(\Delta\Psi_2)$) are constants. Therefore, the AoA measuring errors (including the zenith and azimuth angle measuring errors $\delta\theta$, $\delta\phi$ respectively) can be expressed as simple functions of zenith and azimuth angle. The radial distance is the distance between the MTSP and the receiver, which is denoted as $R_s$. $R_s$ can be determined by combining the AoA, baseline length $d_i$, and the radio wave propagation path length R (Stober and Chau, 2015). The geometry is shown in figure 4(a). $\alpha$ is the angle between the baseline (i.e., axis-$X_i$) and the line from the receiver to the MTSP (denoted as point A). If $\alpha$, $d_i$ and R are known, $R_s$ can be calculated easily using the Cosine Law as:

$$R_s = \frac{R^2 - d_i^2}{2(R - d_i \cos\alpha)} \qquad (1)$$

A multistatic configuration will influence the accuracy of $R_s$ (denoted as $\delta R_s$). This is because $\alpha$, d and R are determined

by the multistatic configuration. We consider the error term $\delta R_s$ in our method, which is ignored in the HM. $\delta R_s$ is a function

of the AoA measuring errors ($\delta\theta$ and $\delta\phi$) and the radio wave propagation path length measuring error (denoted as $\delta R$). $\delta R$

is caused by the measuring error of the wave propagation time $\delta t$, which is approximately $21\mu s$ (Kang, 2008). Thus, $\delta R$ can

be set as a constant and the default value in our program is $\delta R = c\delta t = 6.3km$. It is worth noting that the maximum

unambiguous range for pulse meteor radars is determined by the pulse repetition frequency (PRF) (Hocking et al., 2001;

Holdsworth et al., 2004). For multistatic meteor radars utilizing forward scatter, the maximum unambiguous range is c/PRF

(where c is the speed of light). For the area where R exceeds the maximum unambiguous range, $\delta R$ is set to positive infinity.

## 2.2 Three kinds of coordinate systems and their transformations

To better depict the multistatic system configuration, three kinds of right-hand coordinate systems need to be established as

shown in figure 3. These are $X_0Y_0Z_0$, $X_iY_iZ_i$ and XYZ. $X_0Y_0Z_0$ is the ENU (east-north-up) coordinate system where axis-

$X_0, Y_0, Z_0$ represent the east, north, up directions respectively. Another two coordinate systems are established to facilitate

different error propagations. All types of errors need to be transformed to the ENU coordinate system $X_0Y_0Z_0$ in the end.

Coordinate system XYZ is established to depict the spatial configuration of the receiving array and has its the origin of XYZ

there as shown in figure 3. Axis-Z is collinear with the antenna boresight and perpendicular to the horizontal plane on which

the receiving array lies. Axis-X and axis-Y are collinear with the arms of the two orthogonal antenna arrays. AoAs will be

represented in XYZ for convenience. Inspection of figure 4 indicates that it is convenient to analyse the range information in

a plane that goes through the baseline and the MTSP. Thus, a coordinate system $X_iY_iZ_i$ is established for a transmitter $T_i$. The

coordinate origins of $X_iY_iZ_i$ are all on the receiving array. We stipulate that axis-$X_i$ points to transmitter $i$ ($T_i$). Each pair $T_i$

and receiver $R_X$ constitutes a radar link, which is referred to as $L_i$. The range related information for each $L_i$ will be

calculated in $X_iY_iZ_i$. Different types of errors need to propagate to and be compared in $X_0Y_0Z_0$ which is convenient for

retrieving wind fields.

We stipulate that clockwise rotation satisfies the right-hand corkscrew rule. By rotating $\psi_x^{X,i}$, $\psi_y^{Y,i}$ and $\psi_z^{Z,i}$ about axis-X, Y

and Z, respectively, one can transform XYZ to $X_iY_iZ_i$. It is worth mentioning that $X_iY_iZ_i$ is non-unique because any rotation

about axis-$X_i$ can obtain another satisfactory $X_iY_iZ_i$. Hence, $\psi_x^{X,i}$ can be set to any value. Similarly, by rotating $\psi_x^{i,0}$, $\psi_y^{i,0}$

and $\psi_z^{i,0}$ about axis-X, Y and Z, respectively, one can transform $X_iY_iZ_i$ to $X_0Y_0Z_0$. To realize the coordinate transformation

between these three coordinate systems, a coordinate rotation matrix $A_R(\psi_x, \psi_y, \psi_z)$ is introduced. Using $A_R$, one can

transform the coordinate point or vector presentation from one coordinate system to another. The details of the coordinate

rotation matrix $A_R(\psi_x, \psi_y, \psi_z)$ can be found in **Appendix (A.1)**.

## 2.3 Two types of measuring errors

The analytical method of the spatial resolution for each radar link is the same. The difference between these radar links is only the value of the six coordinate rotation angles ($\psi_x^{X,i}$, $\psi_y^{Y,i}$ and $\psi_z^{Z,i}$; $\psi_x^{i,0}$, $\psi_y^{i,0}$ and $\psi_z^{i,0}$) and the baseline distance $d_i$. The spatial resolution related measurement errors which will cause location errors of the MTSP can be classified into two types: $E_1$ is caused by measurement errors at the receiver, and $E_2$ is due to the pulse length. These two errors are mutually independent. Hence, the total error ($E_{total}$) can be expressed as:

$$E_{total}^2 = E_1^2 + E_2^2 \tag{2}$$

$E_1$ is related to three indirect measuring errors. They are zenith, azimuth and radial distance measuring errors, denoted as $\delta\theta$, $\delta\phi$ and $\delta R_s$ respectively. In XYZ, $E_1$ can be decomposed into three orthogonal error vectors using $\delta\theta$, $\delta\phi$ and $\delta R_s$ (see figure 4(c)) which we now explain in more detail. PDMEs, i.e., $\delta(\Delta\Psi_1)$ and $\delta(\Delta\Psi_2)$, are caused by some practical factors, such as phase calibration mismatch and the fact that the specular point is not actually a point but is a few Fresnel zones in length. A meteor radar system calculates phase differences between different pairs of antennas though cross-correlations and then fits them to get the most likely AoAs. Therefore, the system needs to be assigned a tolerance value of $\delta(\Delta\Psi_1)$ and $\delta(\Delta\Psi_2)$. Different meteor radar systems have different AoA-fit algorithms and thus different AoA measuring error distributions. To analyse the spatial resolution for a SIMO meteor radar system as generally as possible and to avoid tedious error propagation at the receiving array, we start the error propagation from $\delta(\Delta\Psi_1)$ and $\delta(\Delta\Psi_2)$ and set them as constants. AoA measuring errors $\delta\theta$ and $\delta\phi$ can then be expressed as:

$$\delta\theta = \frac{\lambda}{2\pi D_1}\frac{\cos\phi}{\cos\theta}\delta(\Delta\Psi_1) + \frac{\lambda}{2\pi D_2}\frac{\sin\phi}{\cos\theta}\delta(\Delta\Psi_2) \tag{3}$$

$$\delta\phi = \frac{\lambda}{2\pi D_2}\frac{\cos\phi}{\sin\theta}\delta(\Delta\Psi_2) - \frac{\lambda}{2\pi D_1}\frac{\sin\phi}{\sin\theta}\delta(\Delta\Psi_1) \tag{4}$$

where $\lambda$ is the radio wavelength, $D_1$ and $D_2$ are the length of the two orthogonal antenna arms, and $\theta$ and $\phi$ are the zenith angle and the azimuth angle, respectively. The details can be found in **Appendix (A.2).** It is worth noting that $\delta\theta$ and $\delta\phi$ are not mutually independent. The expectation value of their product is not identical to zero unless $\frac{E\left(\delta^2(\Delta\Psi_1)\right)}{D_1^2}$ is equal to $\frac{E\left(\delta^2(\Delta\Psi_2)\right)}{D_2^2}$.

$\delta R_s$ can be expressed as a function of $\delta R$, $\delta\theta$ and $\delta\phi$ as:

$$\delta R_s = F(\delta R, \delta\theta, \delta\phi) = f_R(\theta,\phi)\delta R + f_\theta(\theta,\phi)\delta\theta + f_\phi(\theta,\phi)\delta\phi \tag{5}$$

$f_R(\theta, \phi)$, $f_\theta(\theta, \phi)$ and $f_\phi(\theta, \phi)$ are the weighting functions of $\delta R_s$. The details about the weighting function and deduction can be found in **Appendix (A.3)**. Inspection of figure 4(c) indicates that $E_1$ can be decomposed into three orthogonal error vectors in coordinate XYZ, denoted as $\overrightarrow{\delta R_s}$, $\overrightarrow{R_s \delta \theta}$ and $\overrightarrow{R_s sin\theta \delta \phi}$. These three vectors can be expressed in XYZ as:

$$\overrightarrow{\delta R_s} = \delta R_s(sin\theta cos\phi, sin\theta sin\phi, cos\theta)^T \tag{6}$$

$$\overrightarrow{R_s \delta \theta} = R_s \delta\theta(cos\theta cos\phi, cos\theta sin\phi, -sin\theta)^T \tag{7}$$

$$\overrightarrow{R_s sin\theta \delta \phi} = R_s sin\theta \delta\phi(-sin\phi, cos\phi, 0)^T \tag{8}$$

$E_2$ is related to the radio wave propagation path. A pulse might be reflected anywhere within a pulse length (see figure 4(b)). This causes a location error in the MTSP, represented as an error vector $\overrightarrow{DA}$. D is the median point of the isosceles triangle $\Delta$ABC's side BC. The representation of the error vector $\overrightarrow{DA}$ can be solved in $X_iY_iZ_i$ by using geometrical relationships as:

$$\overrightarrow{DA} = \left( \frac{(2-a_1-a_2)x_i + d_i(a_2-1)}{2}, \frac{(2-a_1-a_2)y_i}{2}, \frac{(2-a_1-a_2)z_i}{2} \right)^T \tag{9}$$

where S is the half pulse length and $a_1 = \frac{R_s - S}{R_s}$. $a_2 = \frac{R_i - S}{R_i}$. $d_i$ is the baseline length. $(x_i, y_i, z_i)$ is the coordinate value of a MTSP (point A in figure 4) in $X_iY_iZ_i$. More details can be found in **Appendix (A4)**

### 2.4 Transformation to ENU coordinates

Thus far, two types of errors in different coordinate systems have been introduced. Now they need to be transformed to ENU coordinates $X_0Y_0Z_0$, in order to compare different radar links and to analyse the wind fields. $E_1$ related error vectors, which are three orthogonal vectors $\overrightarrow{\delta R_s}$, $\overrightarrow{R_s \delta \theta}$ and $\overrightarrow{R_s sin\theta \delta \phi}$ and represented in XYZ as eq.(6)-(8), and need to be transformed from $XYZ$ to $X_0Y_0Z_0$. To project $\overrightarrow{\delta R_s}$, $\overrightarrow{R_s \delta \theta}$ and $\overrightarrow{R_s sin\theta \delta \phi}$ towards axis-$X_0, Y_0, Z_0$ respectively, and reassemble them to form three new error vectors in axis-$X_0, Y_0, Z_0$. Using the coordinate rotation matrix $A_R^{(XYZ, X_0Y_0Z_0)} = A_R(\Psi_x^{i,0}, \Psi_y^{i,0}, \Psi_z^{i,0}) \cdot A_R(\psi_x^{X,i}, \psi_y^{Y,i}, \psi_z^{Z,i})$ and eq.(6)-(8), the unit vectors of those three vectors can be represented in $X_0Y_0Z_0$ as:

$$\begin{pmatrix} X_0'(\delta R_s) & X_0'(\delta\theta) & X_0'(\delta\phi) \\ Y_0'(\delta R_s) & Y_0'(\delta\theta) & Y_0'(\delta\phi) \\ Z_0'(\delta R_s) & Z_0'(\delta\theta) & Z_0'(\delta\phi) \end{pmatrix} = A_R^{(XYZ, X_0Y_0Z_0)} \cdot \begin{pmatrix} sin\theta cos\phi & cos\theta cos\phi & -sin\phi \\ sin\theta sin\phi & cos\theta sin\phi & cos\phi \\ cos\theta & -sin\theta & 0 \end{pmatrix} \tag{10}$$

$(X_0'(\delta R_s), Y_0'(\delta R_s), Z_0'(\delta R_s))^T$, $(X_0'(\delta\theta), Y_0'(\delta\theta), Z_0'(\delta\theta))^T$, $(X_0'(\delta\phi), Y_0'(\delta\phi), Z_0'(\delta\phi))^T$ are unit vectors of $\overrightarrow{\delta R_s}$, $\overrightarrow{R_s \delta \theta}$ and $\overrightarrow{R_s sin\theta \delta \phi}$ in $X_0Y_0Z_0$ respectively. The $3 \times 3$ matrix on the left hand side of the eq.(10) is denoted as $P_{ij}$ for $i, j = 1,2,3$.

From eq.(6)-(8) and figure 4(c), we see that the length of those three vectors (the error values) are $\delta R_s$, $R_s \delta\theta$, $R_s sin\theta \delta\phi$ as a function of $\delta R$, $\delta\theta$, $\delta\phi$. In order to reassemble them to form new error vectors, transformation of $\delta\theta$ and $\delta\phi$ into two independent errors $\delta(\Delta\Psi_1)$ and $\delta(\Delta\Psi_2)$ is needed because $\delta\theta$ and $\delta\phi$ are not independent. Using eq. (3) and (4), one can

transform vector $(\delta R\,,\ \delta\theta\,,\ \delta\phi)^T$ to three independent measuring errors $\delta R$, $\delta(\Delta\Psi_1)$ and $\delta(\Delta\Psi_2)$. And thus $(\delta R_s,\ R_s\delta\theta,\ R_s sin\theta\delta\phi)^{\mathrm{T}}$ can be expressed as:

$$
\begin{pmatrix} \delta R_s \\ R_s\delta\theta \\ R_s sin\theta\delta\phi \end{pmatrix} = \begin{pmatrix} f_R(\theta,\phi) & f_\theta(\theta,\phi) & f_\phi(\theta,\phi) \\ 0 & R_s & 0 \\ 0 & 0 & R_s sin\theta \end{pmatrix} \cdot \begin{pmatrix} 1 & 0 & 0 \\ 0 & \frac{\lambda}{2\pi}\frac{cos\phi}{cos\theta\ D_1} & \frac{\lambda}{2\pi}\frac{sin\phi}{cos\theta\ D_2} \\ 0 & -\frac{\lambda}{2\pi}\frac{sin\phi}{sin\theta\ D_1} & \frac{\lambda}{2\pi}\frac{cos\phi}{sin\theta\ D_2} \end{pmatrix} \cdot \begin{pmatrix} \delta R \\ \delta(\Delta\Psi_1) \\ \delta(\Delta\Psi_2) \end{pmatrix} \quad (11)
$$

The product of the first and the second term on the right hand side of eq.(11) is a $3\times3$ matrix, denoted as $W_{ij}$ for $i,j=$
1,2,3. From eq.(11), we see that the three error values $\delta R_s,\ R_s\delta\theta,\ R_s sin\theta\delta\phi$ are the linear combinations of $\delta R$,
$\delta(\Delta\Psi_1)$, and $\delta(\Delta\Psi_2)$ with their corresponding linear coefficients $W_{1j}, W_{2j}$, and $W_{3j}$. Those three error values can be projected toward new directions (e.g., axis-$X_0, Y_0, Z_0$) by using $P_{ij}$. It worth noting that in a new direction, a same basis's projected linear coefficients from different error values should be used to calculate their sum of squares (SS). And then the square root of SS will be used as a new linear coefficient for that basis in the new direction. For example, in $X_0$ directions, basis $\delta(\Delta\Psi_1)$'s projected linear coefficients are $X_0'(\delta R_s)W_{12}$, $X_0'(\delta\theta)W_{22}$, $X_0'(\delta\phi)W_{32}$ from $\overrightarrow{\delta R_s}$, $\overrightarrow{R_s\delta\theta}$ and $\overrightarrow{R_s sin\theta\delta\phi}$
respectively. Therefore, the new linear coefficient for $\delta(\Delta\Psi_1)$ in the $X_0$ direction is $W_{X_0'}^{\delta(\Delta\Psi_1)}=$
$\pm\sqrt{(X_0'(\delta R_s)W_{12})^2+(X_0'(\delta\theta)W_{22})^2+(X_0'(\delta\phi)W_{32})^2}$. Similarly, one can get $\delta R$ and $\delta(\Delta\Psi_2)$'s new linear coefficients in $X_0'$, denoted as $W_{X_0'}^{\delta R}$ and $W_{X_0'}^{\delta(\Delta\Psi_2)}$. Thus, the true error value in the $X_0$ direction is $W_{X_0'}^{\delta R}\delta R+W_{X_0'}^{\delta(\Delta\Psi_1)}\delta(\Delta\Psi_1)+$
$W_{X_0'}^{\delta(\Delta\Psi_2)}\delta(\Delta\Psi_2)$. Because $\delta R$, $\delta(\Delta\Psi_1)$, and $\delta(\Delta\Psi_2)$ are mutually independent, $E_1$ is related to the mean square error (MSE) values in the $X_0$ direction, denoted as $\delta_{(1)}X_0$ and can be expressed as $\delta_{(1)}X_0=$
$\pm\sqrt{\left(W_{X_0'}^{\delta R}\delta R\right)^2+\left(W_{X_0'}^{\delta(\Delta\Psi_1)}\delta(\Delta\Psi_1)\right)^2+\left(W_{X_0'}^{\delta(\Delta\Psi_2)}\delta(\Delta\Psi_2)\right)^2}$.

In short, $E_1$ related errors in ENU coordinate's three axis directions (denoted as $\delta_{(1)}X_0$, $\delta_{(1)}Y_0$ and $\delta_{(1)}Z_0$) can be expressed in the form of a matrix as:

$$
\begin{pmatrix} \delta_{(1)}^2 X_0 \\ \delta_{(1)}^2 Y_0 \\ \delta_{(1)}^2 Z_0 \end{pmatrix} = P_{ij}^2 \cdot W_{ij}^2 \cdot \begin{pmatrix} \delta^2 R \\ \delta^2(\Delta\Psi_1) \\ \delta^2(\Delta\Psi_2) \end{pmatrix} \quad (12)
$$

*The* $E_2$ related error vector $\overrightarrow{DA}$ needs transformation from $X_iY_iZ_i$ to $X_0Y_0Z_0$. Therefore, $E_2$ related errors in the ENU
coordinate's three axis directions (denoted as $\delta_{(2)}X_0$, $\delta_{(2)}Y_0$ and $\delta_{(2)}Z_0$) can be expressed in the form of a matrix as:

$$
\begin{pmatrix} \delta_{(2)}X_0 \\ \delta_{(2)}Y_0 \\ \delta_{(2)}Z_0 \end{pmatrix} = \pm A_R(\Psi_x^{i,0}, \Psi_y^{i,0}, \Psi_z^{i,0}) \cdot \overrightarrow{DA} \quad (13)
$$

$E_1$ and $E_2$ are mutually independent. By using eq.(1), the total MSE values in ENU coordinate's three axis directions (denoted as $\delta_{total}X_0$, $\delta_{total}Y_0$ and $\delta_{total}Z_0$) can be expressed in the form of matrix as:

$$\begin{pmatrix} \delta_{total}^2 X_0 \\ \delta_{total}^2 Y_0 \\ \delta_{total}^2 Z_0 \end{pmatrix} = \begin{pmatrix} \delta_{(1)}^2 X_0 \\ \delta_{(1)}^2 Y_0 \\ \delta_{(1)}^2 Z_0 \end{pmatrix} + \begin{pmatrix} \delta_{(2)}^2 X_0 \\ \delta_{(2)}^2 Y_0 \\ \delta_{(2)}^2 Z_0 \end{pmatrix} \tag{14}$$

In conclusion, for a radar link $L_i$ and a MTSP represented as $(x_0, y_0, z_0)$ in the ENU coordinate system $X_0Y_0Z_0$, as sketched in figure 4(a), the location errors of this point in east, north and up directions ($\pm\delta_{total}X_0$, $\pm\delta_{total}Y_0$ and $\pm\delta_{total}Z_0$) can be calculated as follows: firstly, for a point $(x_0, y_0, z_0)$ in $X_0'Y_0'Z_0'$, use $A_R$ to transform it to $X_iY_iZ_i$ and denote it as $(x_i, y_i, z_i)$. Then in $X_iY_iZ_i$ calculate the AoA ($\theta$ and $\phi$) and the range information ($R_s$ and $R_i$). Details of AoA and range calculation can be found in **Appendix (A.5)**. It's worth noting that the AoA is given by the angles relative to

the axes of XYZ. Secondly, in XYZ using the AoA and eq.(3)-(8) to calculate $E_1$'s three orthogonal error vectors shown in figure 4(c); in $X_iY_iZ_i$ use the range information and eq.(9) to calculate $E_2$'s error vector $\overrightarrow{DA}$ as shown in figure 4(b). Thirdly, project $E_1$'s three error vectors to $X_0Y_0Z_0$ by using eq.(10) and use eq.(11)-(12) to reassemble them to calculate $E_1$ related MSE values in the direction of $X_0, Y_0, Z_0$; use eq.(13) to transform the $E_2$ error vector from $X_iY_iZ_i$ to $X_0Y_0Z_0$. Finally, use eq. (14) to get the total location errors of a MTSP in $(x_0, y_0, z_0)$. Figure 5(a) shows the flow chart for the

process we have just described.

## 3 Results and Discussion

The program to study the method we have described above is written in the python language and is presented in the supplement. To calculate a special configuration of a multistatic radar system, we initially need to set six coordinate transformation angles ($\psi_x^{X,i}$, $\psi_y^{Y,i}$ and $\psi_z^{Z,i}$; $\psi_x^{i,0}$, $\psi_y^{i,0}$ and $\psi_z^{i,0}$) and the baseline length $\mathbf{d_i}$ for each radar link $L_i$. For example; $\psi_x^{i,0} = \psi_y^{i,0} = 0$,

$\psi_z^{i,0} = 30°$ and $d_i = 250km$ means that transmitter $T_i$ is 250 km, 30° east by south of the receiver $R_X$; Further, $\psi_x^{X,i} = 5°, \psi_y^{Y,i} = 0, \psi_z^{Z,i} = 0$ means one receiver arm (axis-Y) points to east by north 60° with 5° elevation. The detection area of interest for a multistatic meteor radar is usually from 70 km to 110 km in height and around 300km×300km in the horizontal. In our program, this area needs to be divided into a spatial grid for sampling. The default value of the sampling grid length is 1 km in height and 5 km in the meridional and zonal directions, respectively. After selecting the desired settings, the program

steps though the sampling grid nodes and calculates the location errors at each node as described in figure 5(a). Figure 5(b) describes the parameter settings and the transversal calculation process. For a given setting of radar link $L_i$, the program will output the squared values of $E_1$ related, $E_2$ related and total MSE ($E_{total}^2$: $\delta_{total}^2 X_0$, $\delta_{total}^2 Y_0$, $\delta_{total}^2 Z_0$; $E_1^2$: $\delta_{(1)}^2 X_0$, $\delta_{(1)}^2 Y_0$, $\delta_{(1)}^2 Z_0$; $E_2^2$: $\delta_{(2)}^2 X_0$, $\delta_{(2)}^2 Y_0$, $\delta_{(2)}^2 Z_0$). The location errors can be positive or negative and thus the spatial resolutions are twice the absolute value of the location errors. For an example, see figure 5(c). For a detected MTSP represented as

$(x_0, y_0, z_0)$ in $X_0Y_0Z_0$, with $\delta_{\text{total}}^2 X_0$ , $\delta_{total}^2 Y_0$ , $\delta_{total}^2 Z_0$ equal to 25, 16 and 9 $\text{km}^2$, respectively, the actual position of the MTSP could occur in an area which is $\pm 5$ km, $\pm 4$ km, $\pm 3$ km around $(x_0, y_0, z_0)$ with equal probability. Consequently, the zonal, meridional and vertical resolutions are 10 km, 8 km and 6 km respectively.

The HM analyses the vertical resolution (corresponding to $\delta Z_0$ in our paper) in a 2-dimensional vertical section (corresponding to the $X_0Z_0$ plane in our paper). To compare with Hocking's work, $\psi_z^{i,0}$ is set to $180°$, and the other five

coordinate transformation angles are all set to zero with **d** equal to 300 km. The half pulse length S is set to 2 km and $\delta(\Delta\Psi_1)$ to $35°$. Calculating in the $X_0Y_0$ plane only should have degraded our method into Hocking's 2-dimensional analysis method, but doesn't because the HM method ignores $\delta R_s$. In fact, the HM considers only $E_2$ and $\overrightarrow{R_s\delta\theta}$ in the $X_0Y_0$ plane. Consequently, we need to further set $f_R(\theta, \phi)$, $f_\theta(\theta, \phi)$ and $f_\phi(\theta, \phi)$ to be zero. When this is done, our method degrades into the HM. Hocking's results are shown as the absolute value of vertical location error normalized relative to the half pulse

width $|\delta Z_0|/S$. Hereinafter, $|E|/S$ is referred to as the normalized spatial resolution such as $\delta_{(1)}X_0$ and $\delta_{total}Y_0$, where E represents the location errors in a direction. Thus, the spatial resolutions are $2S$ times the normalized spatial resolutions.

Our normalized vertical resolution distributions are shown in figure 6(a) and are the same as those presented in Hocking's work (Hocking, 2018). The distribution of $\overrightarrow{R_s\delta\theta}$ related, $E_2$ related, and total normalized vertical resolution distributions are shown in figure 6 from left to right, respectively. In most cases, $E_2$ is an order of magnitude smaller than $\overrightarrow{R_s\delta\theta}$. Only in

the region directly above the receiver does $E_2$ have the same magnitude as $\overrightarrow{R_s\delta\theta}$. In other words, only in the region directly above the receiver can $E_2$ influence the total resolution. $E_2$ is related to the bistatic configuration, but $\overrightarrow{R_s\delta\theta}$ is not. Therefore, in the HM, the distribution of the total vertical resolution varies slightly with **d**. After adding the error term $\overrightarrow{\delta R_t}$, which is related to the bistatic configuration, the normalized total vertical spatial resolution distribution changes more obviously with **d,** as figure 7's first two rows show. The region between the two black lines represents the sampling volume for the receiver

where the elevation angle is beyond $\mathbf{30°}$. As the transmitter/receiver distance become longer, resolutions in this sampling volume are not always acceptable. In figure 7's first row, the transmitter/receiver distance is 300 km and about half of the region between two black line have normalized vertical resolution values larger than 3 km. Because our analytical method can obtain spatial resolutions in 3-dimensional space, figure 7's third row shows a perspective to the horizontal section at 90 km altitude for figure 7's second row.

To get a perspective on the spatial resolution distribution in 3-dimensional space, figure 8 shows the normalized zonal, meridional and vertical spatial resolution distributions for a multistatic radar link whose transmitter/receiver separation is 180 km and the transmitter is south by east $30°$ of the receiver. The classic monostatic meteor radar is a special case of a multistatic meteor radar system whose baseline length is zero. By setting the transmitter/receiver distance to be zero in our program, a monostatic meteor radar's spatial resolution can also be obtained. In this case, the spatial resolution distributions are highly

symmetrical and correspond to the real characteristics of monostatic meteor radar (this is not shown here, but can be found in supplement SF1). In the discussion above, the receiver and transmitter antennas are all coplanar. By varying $\psi_x^{X,i}$ ,

$\psi_y^{Y,i}$ and $\psi_z^{Z,i}$ in our program, non-coplanar receiver/transmitter-antennas situations can also be studied. Slightly tilting the receiver horizontal plane (for example, set $\psi_x^{X,i}=\psi_y^{Y,i} = 5°$) causes the horizontal spatial distributions to change (see SF2 and SF3 in the supplement). In practice, the Earth's curvature and local topography will lead to tilts in the receiver horizontal plane. This kind of tilt should be taken into account for multistatic meteor radar systems and details relating to the parameter selections for this can be found in the supplement.

The AoA error propagation process has been simplified to yield eq.(3)-(4) by using constant PDMEs. This is for the sake of providing the most general example of our method. If the analysis of AoA errors were to start from the original received voltage signals (e.g., Vaudrin et al., 2018), the error propagation process would depend on the specific receiver interferometer configuration and the specific signal processing method. The approach used here can be applied to different receiver antenna configurations or new signal processing algorithms. This would involve substitution of $\delta(\Delta\Psi_1)$ and $\delta(\Delta\Psi_2)$ into other mutually independent measuring errors to suit the experimental arrangement and then establishing a new AoA error propagation to obtain $\delta\theta$ and $\delta\phi$. This means rewriting the second and third term in eq. (11) to the determine a new AoA error propagation matrix and new mutually independent measuring errors, respectively.

It worth noting that except for using the PDMEs as the start of the error propagation, all the analytical processes are built on mathematical error propagations. PDMEs include uncontrolled errors, such as those resulting from the returned wave being scattered from a few Fresnel zones along the meteor trail, phase calibration inaccuracy, and noise. However, there are other error sources in practice. For example, aircraft or lightning interference and fading clutter from obstacles can cause further measurement errors in the AoA. These issues are related to actual physical situations and beyond the scope of this text.

Knowing the valid observational volume for meteor detections and the errors associated with each detection is vital for a meteor radar system as it determines which meteors can be used to calculate wind velocities and also the uncertainties associated with the winds themselves. To reduce the influence of mutual antenna coupling or ground clutter, the elevation angle of a detection should be above a threshold, and $30°$ is typically used, and this sets the basic valid observational volume. Within this, the normalised vertical resolution varies, and in Figure 7 and SF4 in the supplement, only the areas of normalized vertical resolution with values below 3 km are shown, which we argue represents an acceptable sampling volume. In addition, as the transmitter/receiver distance increases, the sampling volume becomes smaller and the vertical resolution in this volume is reduced. This effect limits the practically usable transmitter/receiver distances for multistatic meteor radars.

The geometry of the multistatic meteor radar case also impacts on the ability of the radar to measure the Doppler shifts associated with drifting meteor trails within the observational volume. This is because the measured Doppler shift is produced by the component of the wind field in the direction of the Bragg Vector, which in the multistatic configuration is divergent from the receiver's line of sight (see e.g., Spargo et al., 2019). The smaller the angle between the Bragg vector and the wind fields, the larger is the Doppler shift (and the higher the SNR). This means that within the observational volume, the angular diversity of the Bragg vector should both be taken into account in the wind retrieval process. A discussion of wind retrievals is beyond the scope of this text and will be considered in future work.

## 4 Conclusion

In this study, we have presented preliminary results from an analytical error method analysis of multistatic meteor radar system measurements of angles of arrival. The method can calculate the spatial resolution (the spatial uncertainty) in the zonal, meridional and vertical directions for an arbitrary receiving antenna array configuration in three-dimensional space. A given detected MTSP is located within the spatial resolution volume with an equal probability. Higher values of spatial resolution mean that this region needs more meteor counts or longer averaging to obtain a reliable accuracy. Our method shows that the spatial configuration of a multistatic system will greatly influence the spatial resolution distribution in ENU coordinates and thus will in turn influence the retrieval accuracy of atmospheric parameters such as the wind field. The multistatic meteor radar system's spatial resolution analysis is a key point in analysing the accuracy of retrieved wind and other parameters. The influence of the spatial resolution on wind retrieval will be discussed in future work.

Multistatic radar systems come in many types, and the work in this paper considers only single-input (single-antenna transmitter) and multi-output (5-antenna interferometric receiver) pulse radar systems. Although the single-input multi-output (SIMO) pulse meteor radar is a classic meteor radar system, other meteor radar systems, such as continuous wave radar systems and MISO (multiple-antenna transmitter and single-antenna receiver) also show good experimental results. Using different types of meteor radar systems to constitute a meteor radar network is the future trend and so we will add the spatial resolution analysis of other system types using our method in the future. We will also validate and apply the spatial resolution analysis in the horizontal wind determination to a multistatic meteor radar system that will be soon be installed in China.

*Code availability.* The program to calculate the 3D spatial resolution distributions is available in the supplement.

*Author contributions*:W.Z, X.X, W.Y designed the study. W.Z deduced the formulas and wrote the program. W.Z wrote the paper for the first version. X.X supervised the work and provided valuable comments. I.M.R revised the paper. All of the authors discussed the results and commented on the paper.

*Competing interest.* The authors declare no conflicts of interests

*Acknowledgements.* This work is supported by the B-type Strategic Priority Program of CAS Grant No. XDB41000000, the National Natural Science Foundation of China (41774158, 41974174, 41831071 and 41904135), the CNSA pre-research Project on Civil Aerospace Technologies No. D020105, the Fundamental Research Funds for the Central Universities, and the Open Research Project of Large Research Infrastructures of CAS "Study on the interaction between low/mid-latitude atmosphere and ionosphere based on the Chinese Meridian Project." WZ thanks Dr Jia Mingjiao for useful discussions and Zeng Jie for checking the equations in the manuscript.

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

**Appendix**

**A.1 Coordinates rotation matrix**

For a right-handed rectangular coordinate system $XYZ$, we rotate clockwise $\Psi_x$ about the axis-x to obtain a new coordinate.

We specify that clockwise rotation satisfies in the right-hand screw rule. A vector in $XYZ$, denoted as $(x, y, z)^T$, is represented as $(x', y', z')^T$ in the new coordinate. The relationship between $(x, y, z)^T$ and $(x', y', z')^T$ is:

$$\begin{pmatrix} x' \\ y' \\ z' \end{pmatrix} = A_x(\psi_x) \begin{pmatrix} x \\ y \\ z \end{pmatrix} = \begin{pmatrix} 1 & 0 & 0 \\ 0 & cos\psi_x & sin\psi_x \\ 0 & -sin\psi_x & cos\psi_x \end{pmatrix} \begin{pmatrix} x \\ y \\ z \end{pmatrix} \tag{A1.1}$$

Similarly, we rotate clockwise $\Psi_y$ about the axis-y to obtain a new coordinate. The presentation for a vector in new coordinates and the original can be linked by a matrix, $A_y(\psi_y)$:

$$A_y(\psi_y) = \begin{pmatrix} cos\psi_y & 0 & -sin\psi_y \\ 0 & 1 & 0 \\ sin\psi_y & 0 & cos\psi_y \end{pmatrix} \tag{A1.2}$$

we rotate clockwise $\Psi_z$ about axis-z to obtain a new coordinate. The presentation for a vector in new coordinates and original can be linked by a matrix $A_z(\psi_z)$:

$$A_z(\psi_z) = \begin{pmatrix} cos\psi_z & sin\psi_z & 0 \\ -sin\psi_z & cos\psi_z & 0 \\ 0 & 0 & 1 \end{pmatrix} \tag{A1.3}$$

For any two coordinate systems $XYZ$ and $X'Y'Z'$ with co-origin, one can always rotate clockwise $\Psi_x$, $\Psi_y$ and $\psi_z$ in order of axis-X, Y, Z respectively, transforming $XYZ$ to $X'Y'Z'$ (figure A.1). The presentation for a vector in $X'Y'Z'$ and $XYZ$ can be linked by a matrix, $A_R(\psi_x, \psi_y, \psi_z)$:

$$A_R(\psi_x, \psi_y, \psi_z) = A_z(\psi_z)A_y(\psi_y)A_x(\psi_x) =$$

$$\begin{pmatrix} \cos\psi_y\cos\psi_z & \sin\psi_x\sin\psi_y\cos\psi_z + \cos\psi_x\sin\psi_z & -\cos\psi_x\sin\psi_y\cos\psi_z + \sin\psi_x\sin\psi_z \\ -\cos\psi_y\sin\psi_z & -\sin\psi_x\sin\psi_y\sin\psi_z + \cos\psi_x\cos\psi_z & \cos\psi_x\sin\psi_y\sin\psi_z + \sin\psi_x\cos\psi_z \\ \sin\psi_y & -\sin\psi_x\cos\psi_y & \cos\psi_x\cos\psi_y \end{pmatrix} \tag{A1.4}$$

We call $A_R(\psi_x, \psi_y, \psi_z)$ the coordinates rotation matrix.

## A.2 AoA measuring errors

In coordinate $XYZ$, AoAs includes zenith angle $\theta$ and azimuth angle $\phi$. In the plane wave approximation, the radio wave is at angle $\gamma_1$ and $\gamma_2$ with an antenna array (figure A.2). There is a phase difference $\Delta\Psi_1$ and $\Delta\Psi_2$ between two antennas (figure 1). See figure 1, $\Delta\Psi_1$ and $\Delta\Psi_2$ can be expressed as:

$$\Delta\Psi_1 = \frac{2\pi D_1 \cos\gamma_1}{\lambda} \tag{A2.1}$$

$$\Delta\Psi_2 = \frac{2\pi D_2 \cos\gamma_2}{\lambda} \tag{A2.2}$$

Using $\gamma_1$, $\gamma_2$ the AoA can be expressed as:

$$\cos^2\gamma_1 + \cos^2\gamma_2 + \cos^2\theta = 1 \tag{A2.3}$$

$$\tan\phi = \frac{\cos\gamma_2}{\cos\gamma_1} \tag{A2.4}$$

Or in another expression:

$$\cos\gamma_1 = \sin\theta\cos\phi \tag{A2.5}$$

$$\cos\gamma_2 = \sin\theta\sin\phi \tag{A2.6}$$

substitute $\cos\gamma_1$ and $\cos\gamma_2$ in (A2.3) and (A2.4) by using (A2.1) and (A2.2):

$$\cos^2\theta = 1 - \left(\frac{\lambda}{2\pi}\right)^2 \left(\frac{\Delta^2\Psi_1}{D_1^2} + \frac{\Delta^2\Psi_2}{D_2^2}\right) \tag{A2.7}$$

$$\ln(\tan\phi) = \ln(D_1\Delta\Psi_2) - \ln(D_2\Delta\Psi_1) \tag{A2.8}$$

(A2.7) and (A2.8) link the phase difference with the AoA and expanding $\theta$ and $\phi$, $\Delta\Psi_1$ and $\Delta\Psi_2$ to first order:

$$2\cos\theta\sin\theta\delta\theta = \left(\frac{\lambda}{2\pi}\right)^2 \left[\frac{2\Delta\Psi_1\delta(\Delta\Psi_1)}{D_1^2} + \frac{2\Delta\Psi_2\delta(\Delta\Psi_2)}{D_2^2}\right] \tag{A2.9}$$

$$\delta\phi = \frac{sin\phi cos\phi}{\Delta\Psi_2}\delta(\Delta\Psi_2) - \frac{sin\phi cos\phi}{\Delta\Psi_1}\delta(\Delta\Psi_1) \tag{A2.10}$$

For (A2.9) and (A2.10), substitute $\Delta\Psi_1$ and $\Delta\Psi_2$ using (A2.1), (A2.2) and (A2.5), (A2.6) to the functions of $\theta$, $\phi$. Now, eq. (3) and eq. (4) have been proven. If the zenith angle $\theta = 0°$, we stipulate that $\frac{cos\phi}{sin\theta}$ and $\frac{sin\phi}{sin\theta}$ are 1.

### A.3 Radial distance measuring error

Expand $R_s$, $R$ and $cos\alpha$ in eq.(1) to first order, $\delta R_s$ can be expressed as a function of $\delta R$ and $\delta(cos\alpha)$:

$$\delta R_s = \frac{R^2 - 2Rd cos\alpha + d^2}{2(R - dcos\alpha)^2}\delta R + \frac{d(R^2 - d^2)}{2(R - dcos\alpha)^2}\delta(cos\alpha) \tag{A3.1}$$

$\alpha$ is the angle between $R_s$ and axis-$X_i$. We denote the zenith and azimuth angles in coordinate-$X_i Y_i Z_i$ as $\theta'$ and $\phi'$, respectively. And the relationship between $\alpha$ and $\theta'$, $\phi'$ is

$$cos\alpha = sin\theta' cos\phi' \tag{A3.2}$$

Using coordinates rotation matrix $A_R(\psi_x^{X,i}, \psi_y^{Y,i}, \psi_z^{Z,i})$, $sin\theta' cos\phi'$ can be expressed as the function of AoA:

$$sin\theta' cos\phi' = A_{11} sin\theta cos\phi + A_{12} sin\theta sin\phi + A_{13} cos\theta$$
$$\tag{A3.3}$$

$A_{ij}$ are represent the elements in matrix $A_R(\psi_x^{X,i}, \psi_y^{Y,i}, \psi_z^{Z,i})$ for $i, j = 1,2,3$.

Using (A3.2) and (A3.3), $\delta(cos\alpha)$ can be expressed as a function of $\delta\theta$ and $\delta\phi$ as:

$$\delta(cos\alpha) = (A_{11} cos\theta cos\phi + A_{12} cos\theta sin\phi - A_{13} sin\theta)\delta\theta + (-A_{11} sin\theta sin\phi + A_{12} sin\theta cos\phi)\delta\phi \tag{A3.4}$$

Finally, $\delta R_s$ can be expressed as the function of $\delta R, \delta\theta, \delta\phi$ as:

$$\delta R_s = F(\delta R, \delta\theta, \delta\phi) = f_R(\theta, \phi)\delta R + f_\theta(\theta, \phi)\delta\theta + f_\phi(\theta, \phi)\delta\phi \tag{A3.5}$$

For:

$$f_R(\theta, \phi) = \frac{d^2 + R^2 - 2Rd(A_{11} sin\theta cos\phi + A_{12} sin\theta sin\phi + A_{13} cos\theta)}{2[R - d(A_{11} sin\theta cos\phi + A_{12} sin\theta sin\phi + A_{13} cos\theta)]^2} \tag{A3.6}$$

$$f_\theta(\theta, \phi) = \frac{d(R^2 - d^2)(A_{11} cos\theta cos\phi + A_{12} cos\theta sin\phi - A_{13} sin\theta)}{2[R - d(A_{11} sin\theta cos\phi + A_{12} sin\theta sin\phi + A_{13} cos\theta)]^2} \tag{A3.7}$$

$$f_\phi(\theta, \phi) = \frac{d(R^2 - d^2)(-A_{11} sin\theta sin\phi + A_{12} sin\theta cos\phi)}{2[R - d(A_{11} sin\theta cos\phi + A_{12} sin\theta sin\phi + A_{13} cos\theta)]^2} \tag{A3.8}$$

## A.4 True error of $E_2$

See figure 4 (b); the total length of side AC and side AB represents the pulse width. Side AC equals side CB and they are both
equal to half of the pulse width S. In $X_iY_iZ_i$, the presentation of point A is $(x_i, y_i, z_i)$, the receiver is $(0,0,0)$ and $T_i$ is $(d,0,0)$.

The distance between $T_i$ and A is $R_i = R - R_s$. We denote that the presentation of point B and C in $X_iY_iZ_i$ as $(x_B, y_B, z_B)$ and $(x_C, y_C, z_C)$, respectively. We use vector collinear to establish equations for B and C. Therefore, one can obtain the coordinates of point B and C by the following equations:

$$(x_B, y_B, z_B)^T = \frac{R_s - S}{R_s} \ (x_i, y_i, z_i)^T \tag{A4.1}$$

$$(x_C - d, y_C, z_C)^T = \frac{R_i - S}{R_i} \ (x_i - d, y_i, z_i)^T \tag{A4.2}$$

For isosceles triangle ABC, the perpendicular line AD intersects side CB at the midpoint D. Then, we obtain the coordinate value of D in $X_iY_iZ_i$ as:

$$(x_D, y_D, z_D) = \frac{1}{2}(x_B + x_c, y_B + y_c, z_b + z_c) = \frac{1}{2}((a_1 + a_2)x_i - a_2d + d, (a_1 + a_2)y_i, (a_1 + a_2)z_i) \tag{A4.3}$$

We denote $a_1 = \frac{R_s - S}{R_s}$, $a_2 = \frac{R_i - S}{R_i}$. Finally, one can obtain the error vector of $E_2$ as vector $\overrightarrow{DA}$ in $X_iY_iZ_i$:

$$\overrightarrow{DA} = \left( \frac{(2 - a_1 - a_2)x_i + d(a_2 - 1)}{2}, \frac{2 - a_1 - a_2}{2}y_i, \frac{2 - a_1 - a_2}{2}z_i \right)^T \tag{A4.4}$$

## A.5 Calculate AoA and range information in $X_iY_iZ_i$

For a space point $(x_i, y_i, z_i)$ in $X_iY_iZ_i$ which represents a MTSP, $R_s$ can be solved easily as:

$$\overrightarrow{R_s} = (x_i, y_i, z_i)$$
$$R_s = \sqrt{x_i^2 + y_i^2 + z_i^2} \tag{A6.1}$$

The distance between transmitter $T_i$ and receiver $R_X$ is $d_i$ as shown in figure 4(a). Thus, the coordinate value of $T_i$ in $X_iY_iZ_i$ is $(d_i, 0,0)$ and $R_i$ can be solved as:

$$R_i = \sqrt{(x_i - d_i)^2 + y_i^2 + z_i^2} \tag{A6.2}$$

Before we calculate the AoAs in $X_iY_iZ_i$, the representation of unit vectors of axis-X, Y, Z in $X_iY_iZ_i$ needs to be known. In
XYZ those unit vectors are easily represented as $(1,0,0)^T$, $(0,1,0)^T$, $(0,0,1)^T$. Though the coordinate rotation matrix $A_R(\psi_x^{X,i}, \psi_y^{Y,i}, \psi_z^{Z,i})$, one can get those unit vector's representation in $X_iY_iZ_i$ as:

$$\overrightarrow{n_x} = (A_{11}, A_{21}, A_{31})^T$$

$$\overrightarrow{n_y} = (A_{12}, A_{22}, A_{32})^T$$

$$\overrightarrow{n_z} = (A_{13}, A_{23}, A_{33})^T \tag{A6.3}$$

$\overrightarrow{n_x}$, $\overrightarrow{n_y}$ and $\overrightarrow{n_z}$ are unit vectors of Axis-X, Y, Z respectively, and $A_{ij}$ are the elements a $3 \times 3$ matrix $A_R(\psi_x^{X,i}, \psi_y^{Y,i}, \psi_z^{Z,i})$

for $i, j = 1,2,3$. Now the AoA can be obtained as:

$$\cos\theta = \frac{\overrightarrow{R_s}}{R_s} \cdot \overrightarrow{n_z} \tag{A6.4}$$

$$\sin\theta = \sqrt{1 - \cos^2\theta}$$

(A6.5)

$$\cos\phi = \frac{\overrightarrow{R_s}}{R_s} \cdot \frac{\overrightarrow{n_x}}{\sin\theta}$$

(A6.6)

$$\sin\phi = \frac{\overrightarrow{R_s}}{R_s} \cdot \frac{\overrightarrow{n_y}}{\sin\theta} \tag{A6.7}$$

For $0° < \theta < 180°$ and $0° \leq \phi < 360°$. When $\theta = 0°$ , we handle it as same as in **Appendix (A.2)**.


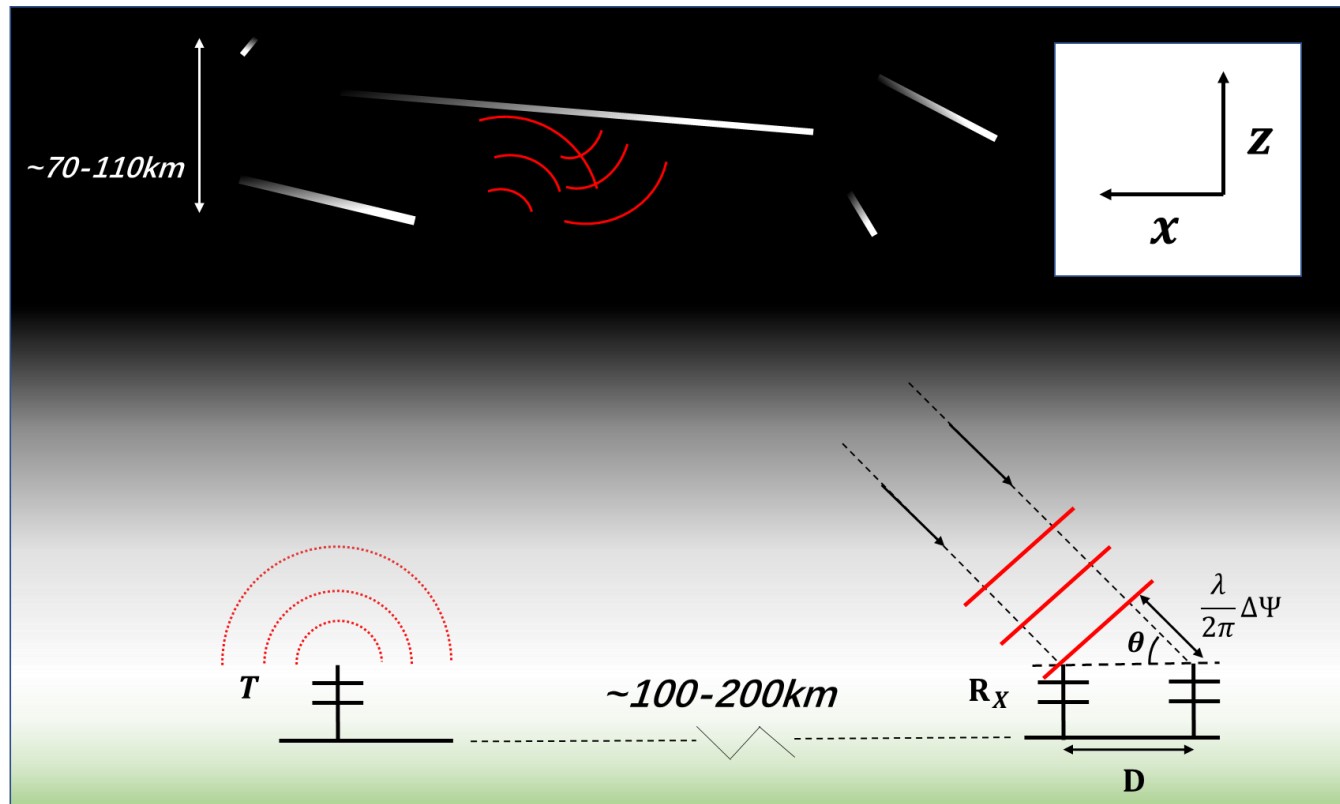

**Figure 1:Schematic diagram of the simplified bistatic configuration used in Hocking's vertical resolution analysis (Hocking, 2018).** **The two receiving antennas and the transmitting antenna are collinear. The analysis is in a 2-dimensional vertical section through the baseline joining the antennas. The radio wave is scattered from a few Fresnel zones of several kilometres' length around the specular point on the meteor trail and received by the receiving antennas. The cross-correlation analysis between the receiving antennas can be used to solve for the AoAs. Because the radio wave is reflected from a region a few Fennel zones in length the measured phase difference between the receiver antenna pairs to deviates from the ideal phase difference. This deviation from the ideal phase difference is one of the error sources in the PDME. In this work, we solve for the ideal phase difference associated with the AoA directed to the MTSP.**

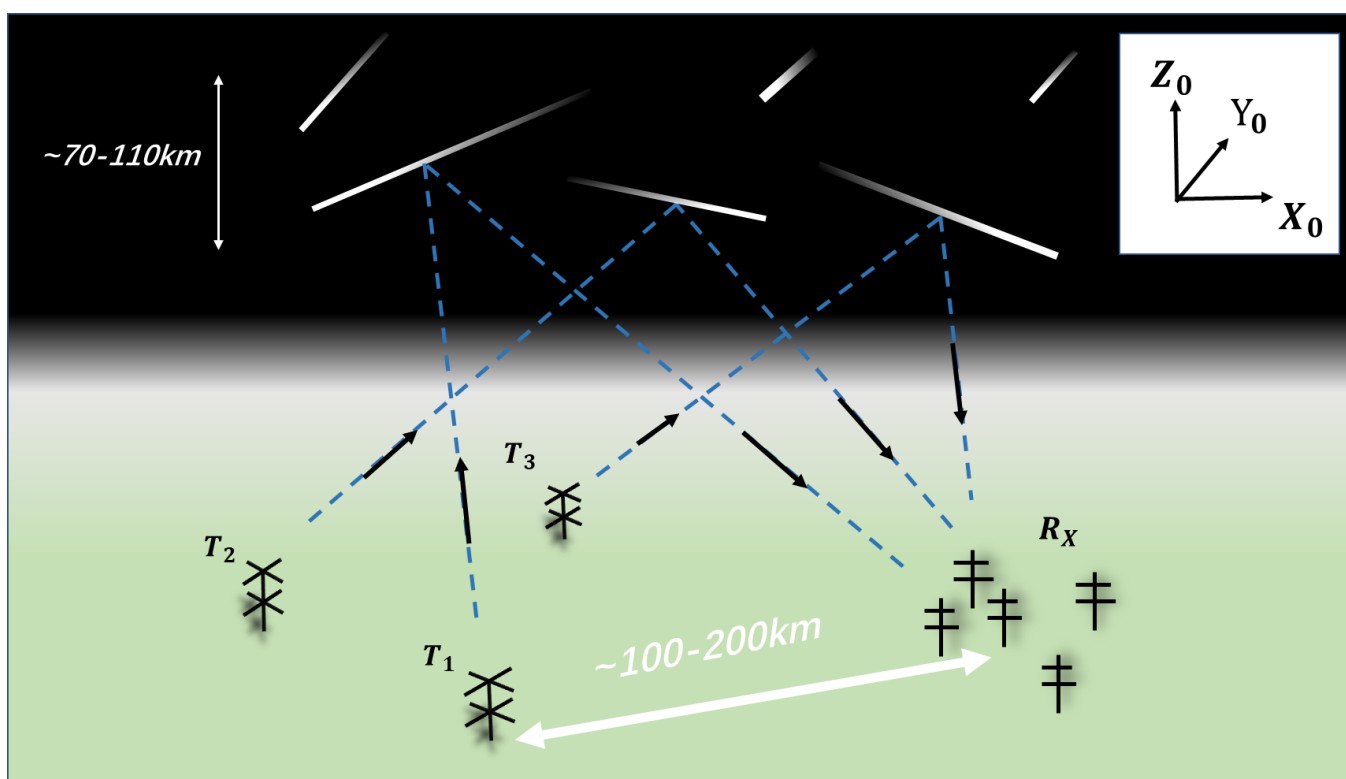

**Figure 2: Schematic diagram of a multistatic meteor radar system using SIMO (single-input and multi-output). There are three transmitters ($T_1, T_2$ $and$ $T_3$) and one receiver ($R_X$) in the picture. The transmitter/receiver distance is typically 100-200 km. $X_0$, $Y_0$, $Z_0$ represents the east, north and up directions of the receiving antenna. Over 90% of the received energy comes from about one kilometre around the specular point of the meteor trail, which is slightly less than the length of the central Fresnel zone (Ceplecha et al., 1998).**

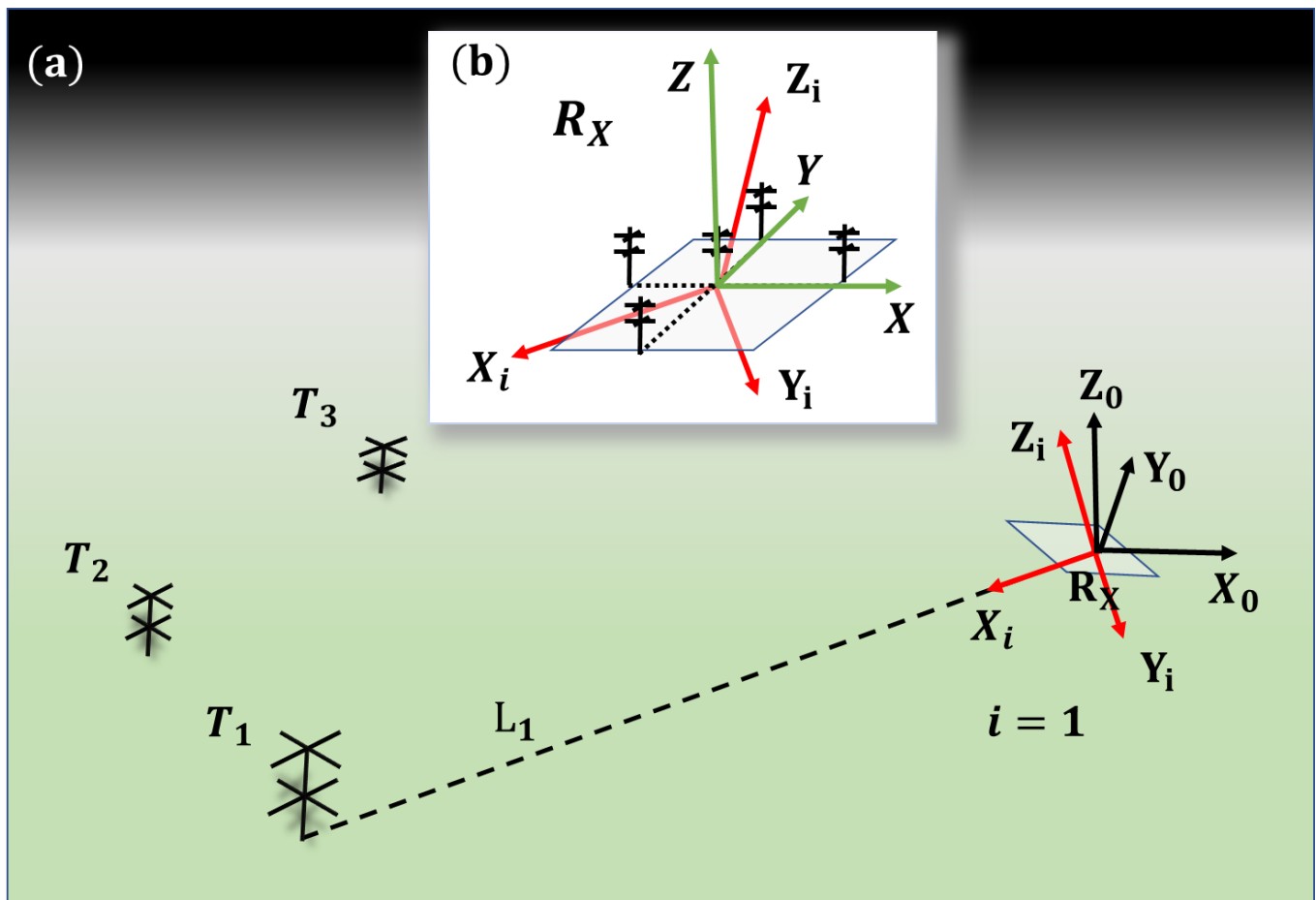

Figure 3: (a) Schematic diagram of the three coordinate systems used in this work. $X_i Y_i Z_i$ are a class of coordinate systems whose axis-$X_i$ points to transmitter I, with, i = 1,2,3. $X_0 Y_0 Z_0$ is the ENU coordinate system to which all errors are compared. (b) Magnified plot of the receiving array. $XYZ$ is fixed on the receiver horizontal plane. Axis-X and Y are collinear with the two arms of the antenna array.

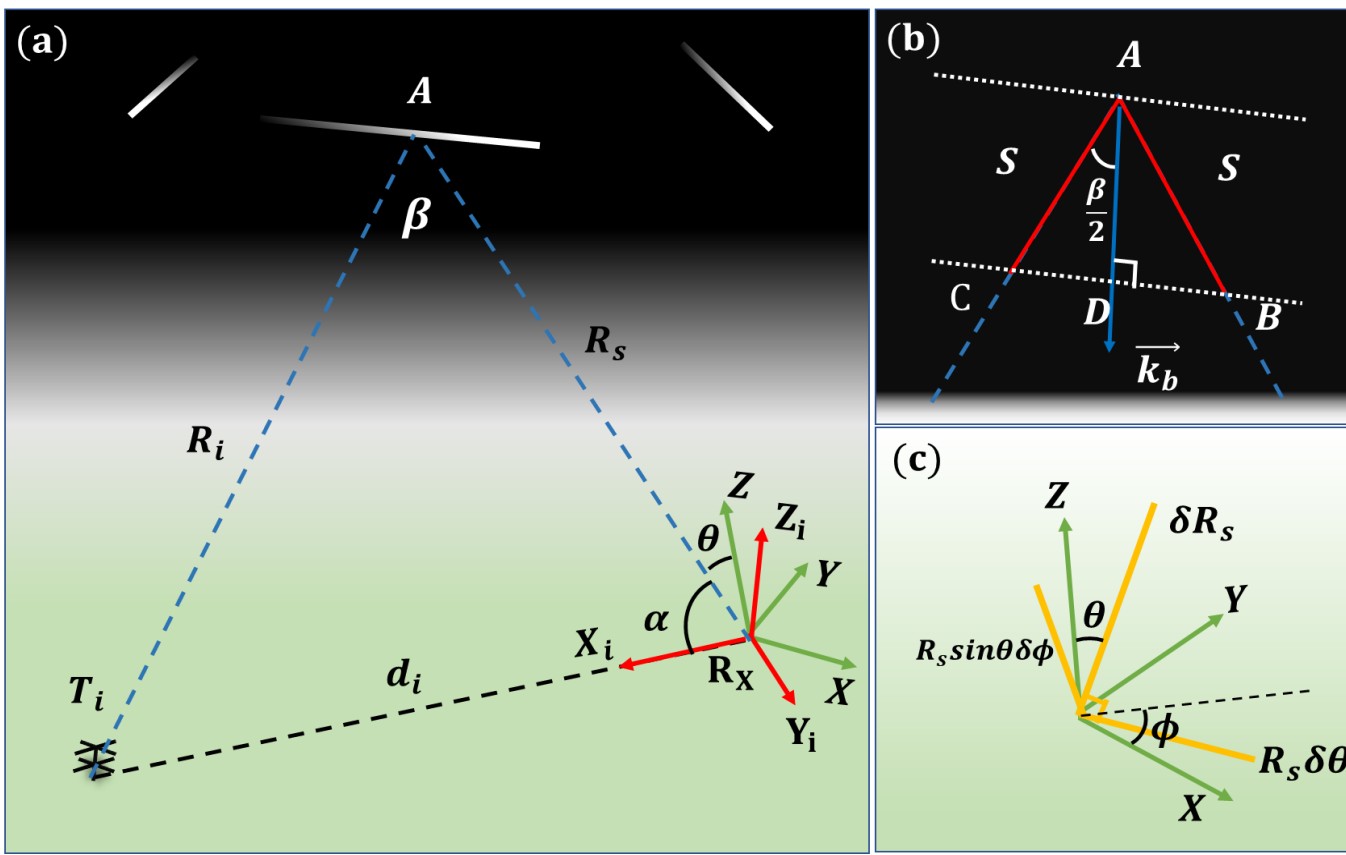

Figure 4: (a) Schematic diagram of the forward scatter geometry for the radar link between $T_i$ and $R_X$. Point-A is the MTSP. (b) Magnified plot of specular point A. The red line represents a radio wave pulse, and S is the half pulse length. $\overrightarrow{k_b}$ is the Bragg vector which halves the forward scatter angle $\beta$. (c) Schematic diagram of $E_1$ in $XYZ$, which can be decomposed into three orthogonal vectors.

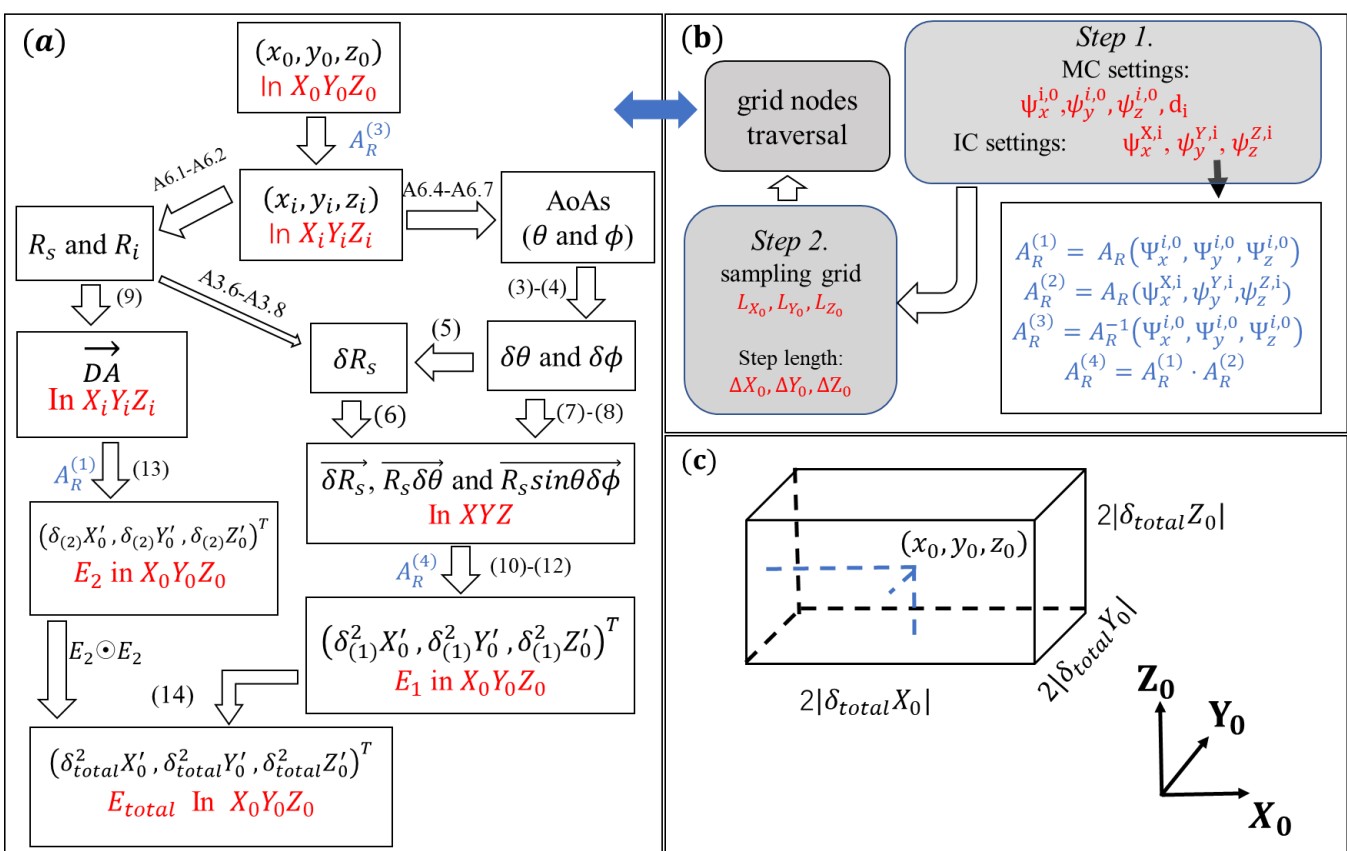

Figure 5: (a) the flow chart of the location error calculation process for a point in $X_0Y_0Z_0$. The notation beside arrows represent the corresponding equations (black) or coordinate rotation matrix (blue) in the paper. "$\odot$" is the Hadamard product. Thus $E_2 \odot E_2$ will yield $(\delta^2_{(2)}X_0, \delta^2_{(2)}Y_0, \delta^2_{(2)}Z_0)^T$. (b) the flow chart of the program to calculate the location errors distributions for a radar link $L_i$. This process includes parameters settings for a radar link; the generation of the sampling grid nodes and the traversing of all the nodes. For each node, the program uses the calculation method described in (a). MC is the multistatic configuration, IC is the interferometer (receiving antenna) configuration. (c) Schematic diagram of the relationship between the spatial resolution and the total location errors of the MTSP. For a detected point in space, the MSE of MTSP's location errors is $\pm|\delta_{total}X_0|$, $\pm|\delta_{total}Y_0|$, $\pm|\delta_{total}Z_0|$ in the zonal, meridional and vertical directions, respectively. This means that the actual specular point might occur in a region which forms a $2|\delta_{total}X_0| \times 2|\delta_{total}Y_0| \times 2|\delta_{total}Z_0|$ cube and the detected point is on the centroid of this cube.

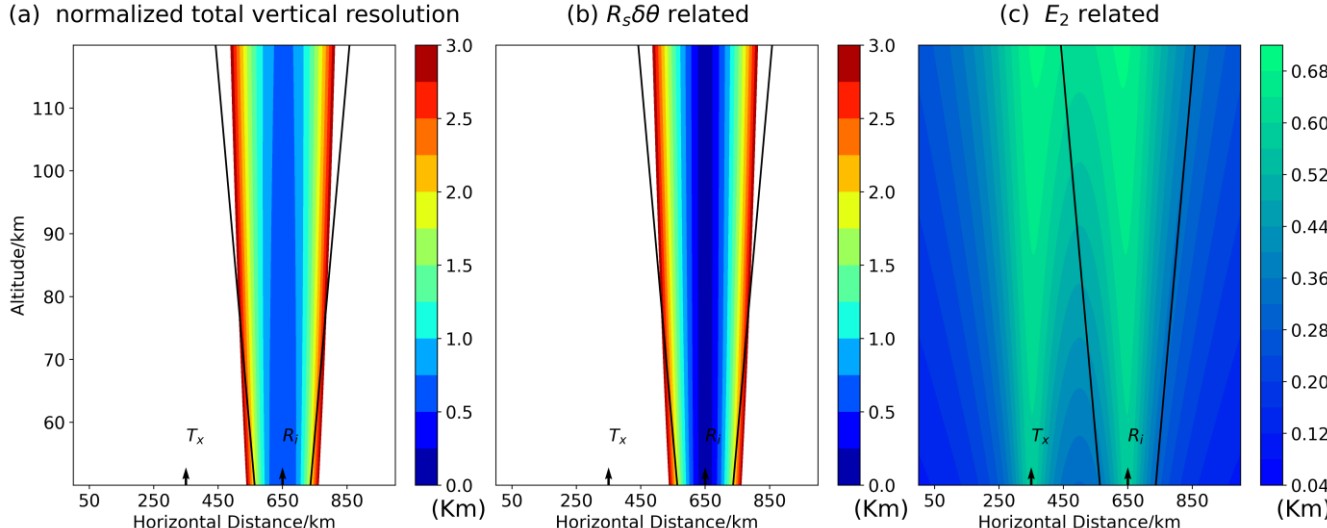

**Figure 6:** the normalized vertical resolution distribution in a vertical section from 50 km to 120 km height when the error term "$\delta R_s$" is ignored. Panels (a), (b), and (c) are the total, $R_s\delta\theta$ related, and $E_2$ related normalized resolution distributions, respectively. These results are the same as those produced in Hocking's work (Hocking, 2018). The two black arrows represent the positions right above the transmitter (Tx) and the receiver (Rx) and the transmitter/receiver separation is 300 km. The region between the two black oblique lines is the sampling volume for the receiving array because the elevation angle is beyond 30° to reduce influence from potential mutual antenna coupling or from other obstacles in the surrounding area. Except the region at large elevation angles (i.e., 90°), the $E_2$ related resolution values are much lower than the $R_s\delta\theta$ related errors. The $R_s\delta\theta$ related resolution distribution depends only on the receiving antennas. Thus, the total vertical resolution distribution is nearly unchanged with the variation of the transmitter/receiver distance. Normalized resolution values that exceed 3 km (which correspond 12 km vertical resolution) are not shown.

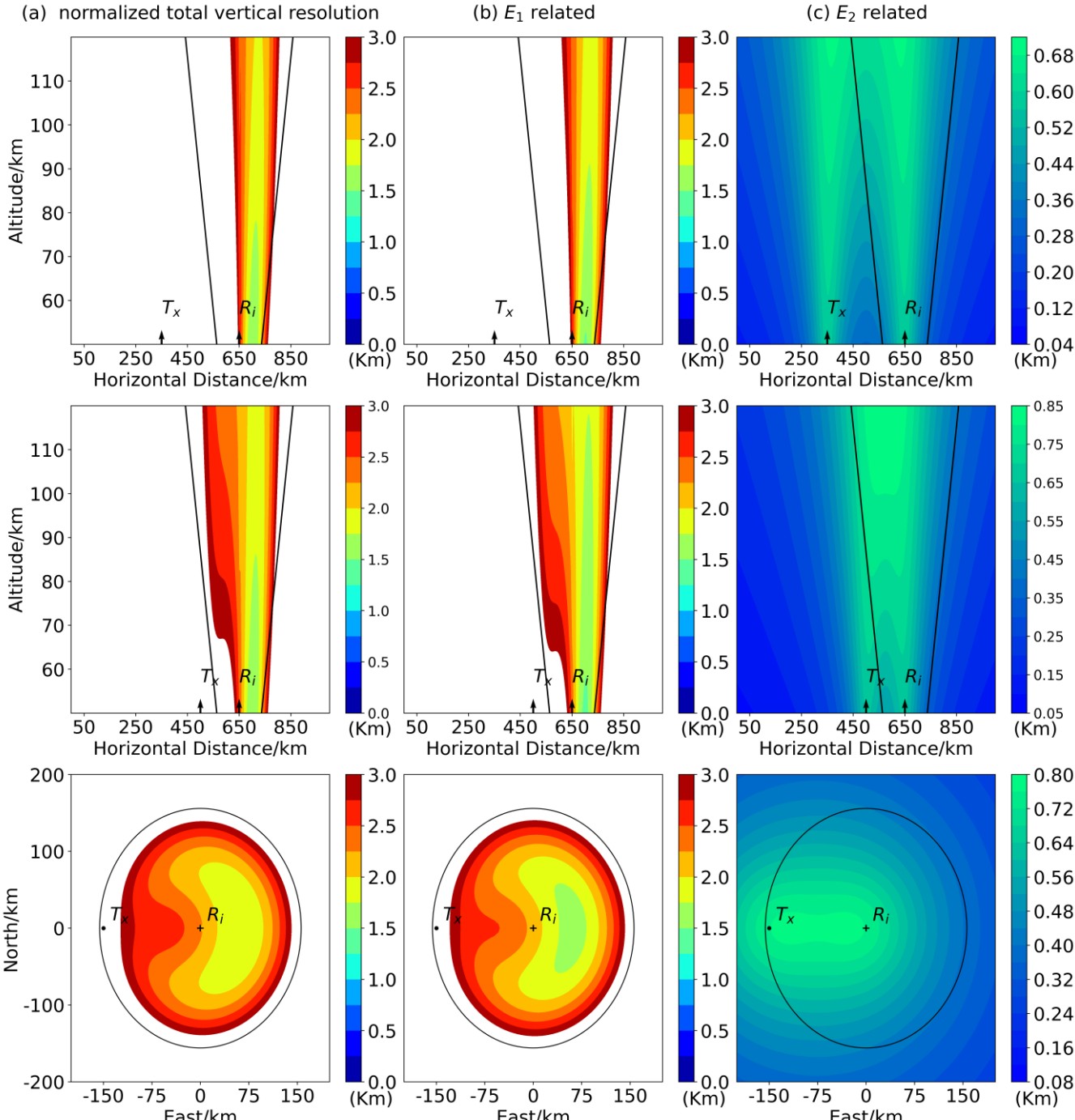

**Figure 7:the normalized vertical resolution distribution using the analytical method described in this paper. The first and second rows represent a vertical section of height from 50 km to 120 km. The third row represents the horizontal section at 90 km and the receiving array is at the origin with positive coordinate values representing the eastward or northward directions, respectively. The first row has the same parameters settings as Figure 6 and is used to compare with Figure 6. The $E_1$ related resolution will change with the transmitter/receiver configuration because it considers the error term "$\delta R_s$". Thus, the total vertical resolution will change**

with the transmitter/receiver configuration. With the transmitter/receiver distance varying from 300 km (the first row) to 150 km (the second row), the total vertical resolution distribution is clearly changed. The third row is the perspective to the horizontal section at 90 km altitude for the second row. Normalized resolution values that exceed 3 km are not shown.

(a)Zonal direction  (b)Meridian direction  (c)Vertical direction

**Figure 8:** the 3D contour plot of the normalized resolution distribution for a multistatic radar link whose baseline length is 180 km and whose transmitter is south by east 30° of the receiver. The black dots represent the position right above the transmitter and the receiving array is at the origin of the axes. (a), (b) and (c) are the normalized resolution distributions in the zonal, meridional and vertical directions, respectively. The subplot's four slice circles from bottom to top are the horizontal section in 50 km, 70 km, 90 km and 110 km height, respectively. The region whose elevation angle of the receiver is less than 30° is not shown and therefore the slice circles become larger from the bottom to the top. Normalized resolution values that exceed 4 km (which corresponds to 16 km resolution) are not shown.

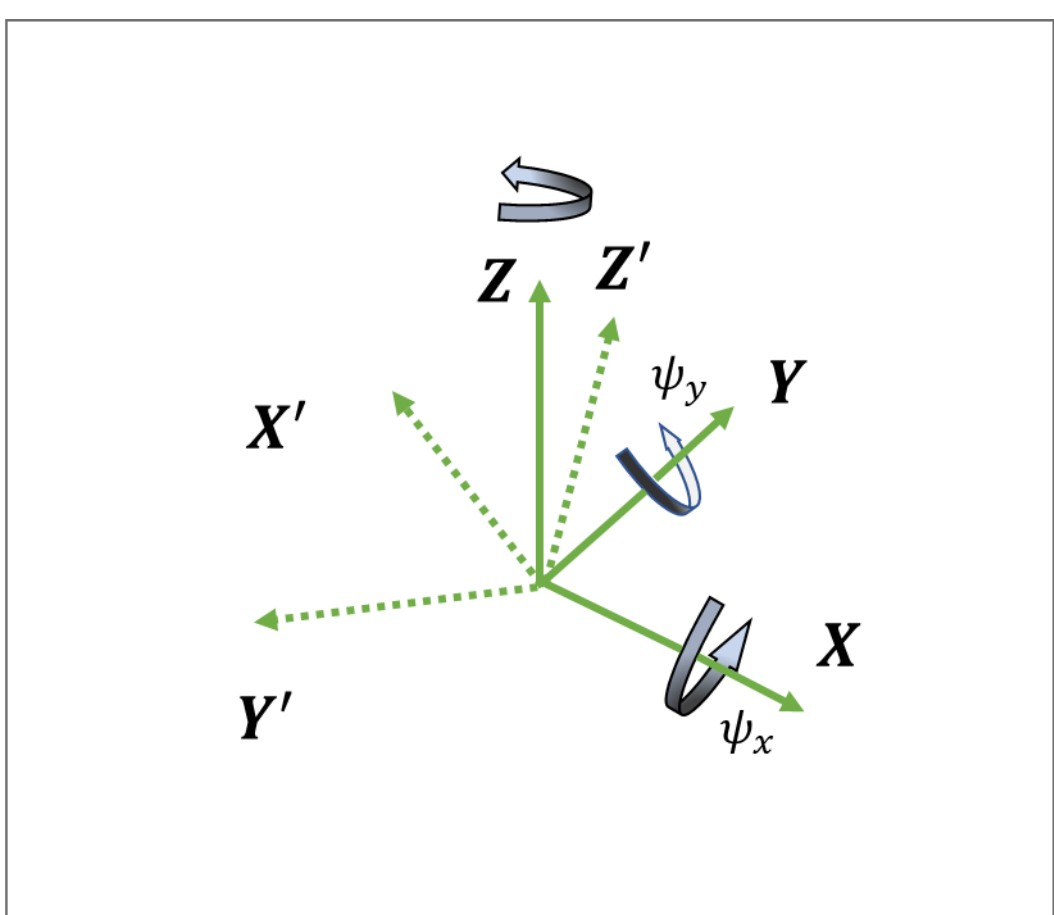

**Figure A.1**

635

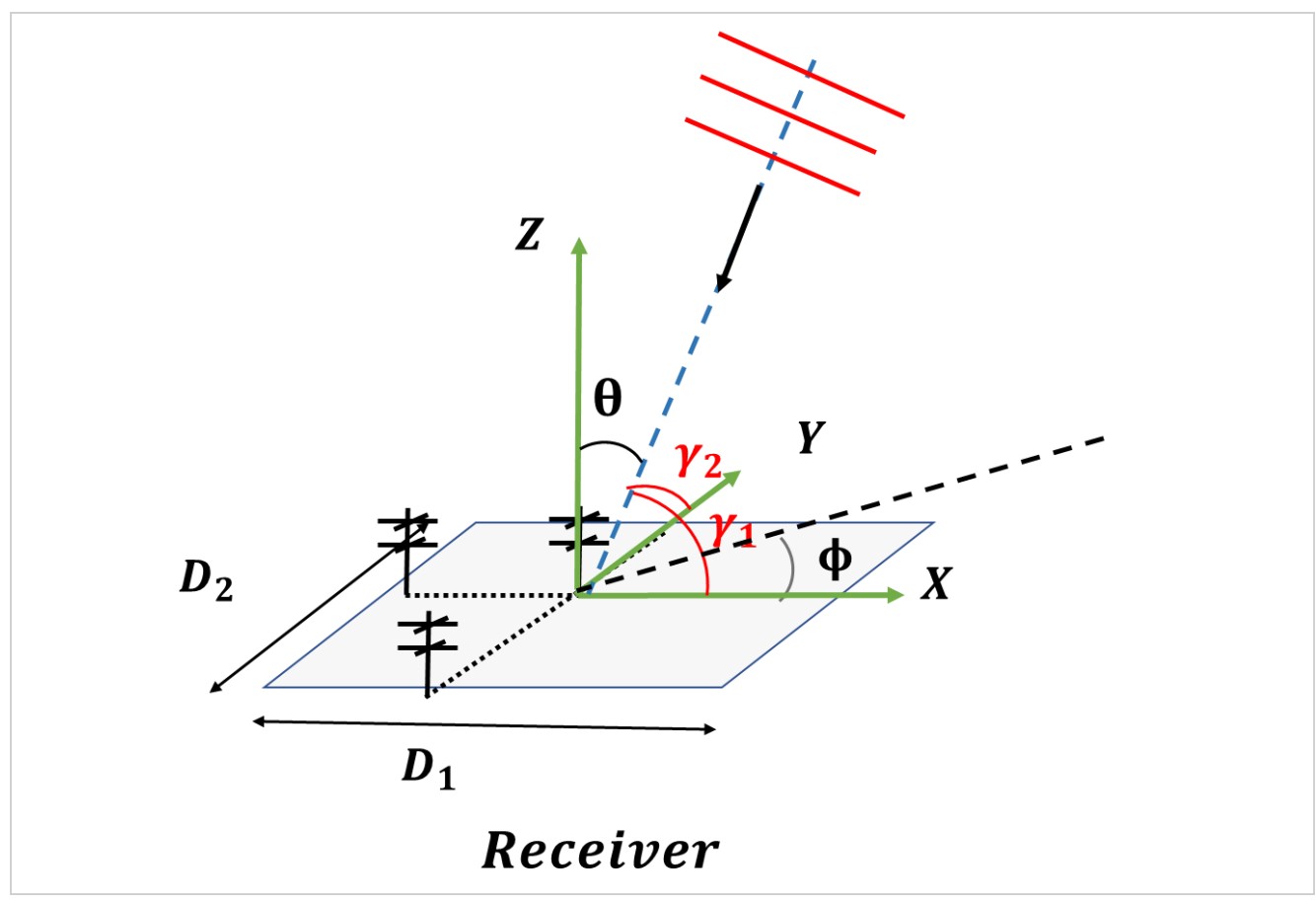

**Figure A.2 The receiving array geometry (only three antennas are shown for clarity)**