# Peer review of "Error analyses of a multistatic meteor radar system to obtain a 3-dimensional spatial resolution distribution"

_Atmospheric Measurement Techniques, 2020_

## Referee Comment (RC1) · Anonymous Referee #1 · 19 Nov 2020

Error analyses of a multistatic meteor radar system to obtain a 3-dimensional spatial resolution distribution

Wei Zhong, Xianghui Xue, Wen Yi, Iain M. Reid, Tingdi Chen, Xiankang Dou

The manuscript presents a theoretical error analysis for statistical multistatic radar systems. Such radars have been known and have been used since decades, but are fairly new for scientific data of meteors and mesosphere/lower thermosphere winds. The basic concept and the related technical details are well-described in textbooks about radar theory. The manuscript shows the algebraic solution of the error propagation considering the uncertainties of the angle of arrival and the sampling or pulse width.

[Figure]

This study is entirely driven by the theoretical aspect on these errors, which might affect the wind retrievals. The authors don't show observations or data. The content of the paper is suitable for a publication at AMT. A proper error propagation is of high relevance and should always been part of a scientific analysis. However, these details presented in the manuscript are often not described in publications.

The reviewer has a few comments that are worth to be included in a revised version.

Major concern:

The reviewer had difficulties to follow some part of the error propagation due to the various introduced coordinate systems denoted as prime without prime and so forth. This made the manuscript very hard to read and one gets easily lost. Although there are some schematics outlining the coordinate systems the reviewer was not able to follow what actually is shown in Figure 5-7. The reviewer was not able to understand the plots reading the figure caption or the corresponding passage in the text. So please describe the color bars in the text or in the caption what they actually mean. The reviewer understands that the authors intended to keep things as general as possible, but some units or quantitative expressions are helpful. In particular, section 2 after line 140 is very hard to read and to follow. This is also partly the case as the Figures are only found at the end of manuscript and one has always to scroll forth and back.

Error budget:

Another important point that should be discussed is that the algebraic errors are just one source that plays a role. The authors should mention in the discussion that there are other error sources as well, originating from the scattering itself or from the experimental set up due to a potential mutual antenna coupling or other obstacles in the surrounding. The later one introduces further biases in the measurements as the angle of arrivals can be significantly altered. Usually HFSS simulation are required to investigate actually the limits of trustworthiness for the interferometry. Furthermore, the authors should mention in the discussion that the scattering occurs not really at

a singular point. The radio wave is bounced back from at least a few Fresnel zones of several kilometer length along the trajectory, which is actually most relevant for the altitude resolution as the radar signal is scattered from an extended volume (1D) and, thus, probes a volume.

Error budget spatial sampling and wind retrieval aspects:

It is also worth to mention and discuss the issues of the sampling volume in the context of the trustworthiness of the interferometry. The schematic in Figure 2 provides a nice example of a multistatic geometry resulting in a less good measurement response compared to a monostatic radar of the same measurement volume, although the set up appears to have a multistatic geometry. The measurement response provides a measure of how well a bragg vector can be inverted to still derive reliable wind speeds (u,v,w). Ideally, all three variables can be estimated with similar measurement response, otherwise biases in one of the wind components are not avoidable. The receiver array in Figure 2 defines the sampling volume. Meteors below a certain elevation angle have to be excluded from the analysis due to the mutual antenna coupling or other ground obstacles causing issues in the interferometry. Further, it is obvious that the angular diversity of the three links inside the remaining sampling volume is less diverse (all are located in a certain sector relative to the receiver) than a monostatic radar and could systematic bias the wind retrievals. This is the nature of the forward scatter ellipse. As all three forward scatter ellipses have the receiver site in the one of their foci points and the bragg vectors always points towards a point along the distance vector between Rx and Tx. It is further obvious that the longer the total path Rt+Rx becomes the less spatial diversity these vectors have, or with other words – all three links start to see the same geometry as it would be the case for a monostatic radar. However, building three receiver sites and using one transmitter would increase the sampling volume and if well-distributed compensates some of this sampling effect on the wind analysis (at least partially), but still has a less good measurement response compared to a monostatic system. I suggest that they add in Figure 5-8 a line or

shading area indicating the angular limit of the receiver/transmitter array by using a truncation elevation angle of about maybe 30°. The actual limit depends on the array set up.
* * *

---

## Referee Comment (RC2) · Anonymous Referee #2 · 3 Dec 2020

**Error analyses of a multistatic meteor radar system to obtain a 3-dimensional spatial resolution distribution**

By W. Zhong, X. Xue, et al.

Submitted to Atmospheric Measurements Techniques (AMT).

This paper presents a study on the error propagation of the measurement errors from multistatic meteor radar systems. Error terms previously not considered, such as the error introduced in the range determination by the geometry of the system configuration, are taken into account in this work. The study focuses only on the theoretical side and uses no real measurements. The topic is of relevance for the community, and AMT constitutes a proper journal for publication of this work. However, this reviewer has some concerns that should be addressed before publication.

**Major comments**

 The manuscript is very difficult to follow, and the English writing needs some improvement. Using the term "wind fields" when referring to monostatic systems is not correct. Monostatic meteor radars can be used to retrieve a mean wind vector, but not wind fields. To obtain the latter, one needs to solve for the gradients. And [du/dy], [dv/dx] can only be estimated when at least a bi-static configuration is taken into account. In connection with this issue, please re-write lines 35-50. Even with a good azimuthal sampling, the shearing term (besides the vorticity) cannot be estimated using a monostatic system. Only [du/dx], [dv/dy] can be estimated from monostatic measurements, but not [du/dy], [dv/dx]. The latter means that not only the vorticity cannot be obtained, but neither the shearing term. Besides, there is no need to have a measure of the vertical wind in order to estimate the horizontal divergence.

Instead of referring to a previous work, it would better if the authors included a simple sketch in order to understand how equation (1) is obtained. Furthermore, the algebraic deductions of the error propagation matrixes presented in the appendixes should be treated with more care. For example, in appendix A.2, it would be helpful to have clearly indicated in its corresponding figure the angles  $\gamma_1$ ,  $\gamma_2$ ,  $\theta$ , and  $\varphi$ . This would help to understand, e.g., how equations A2.3 and A2.4 are obtained. In the case of appendix A.1, please modify its corresponding figure. Since the authors use left-handed coordinate systems but follow the right-hand corkscrew rule, figure A.1 in its present form does not help to understand appendix A.1.

Figures 5 to 8 contain the most important results of this work but they are poorly described and barely discussed. Besides, some of the statements based on these figures are not evident, at least for this reviewer. For example, what is stated in lines 225-226 is not obvious for the eyes of this reviewer.

- 2) This reviewer understands that the authors' objective is to analyse the errors that result from the multistatic configuration. However, the existence of other errors should be mentioned in the paper and a brief discussion on how they compare to the errors here analysed should be included. For example, it is known that the echoes do not originate on a single point in space. So, how large would it be the impact of this on the vertical resolution? Or can it be neglected?
- 3) Maybe it is out of the scope of this work, but it would be helpful if some data were considered in the study. For example, what does really mean having a spatial resolution of let us say, 2-3 km? How would this impact on winds and horizontal gradients estimates? Have the authors made any rough estimation of this? It would be very useful for the readers if some information on this was included in the manuscript.

**Minor comments**

Line 30: please include more references here. Studies from other scientific institutions, e.g., Leipzig University and the Leibniz-IAP (Germany), which have long traditions on studies based on meteor radar measurements should be included.

Line 32: please change "... same height range be processed..." to "... same height range **are** processed..."

Line 48: "Even by releasing...". I think the authors meant "relaxing".

Lines 53 and 59: it is MMARIA, not MMARA. Please change that.

Line 62: it should be "... Chau et al. used two adjacent..." and not "Stober et al."

Lines 65-66, what do the authors mean with "meteor radar data processing method"?

Lines 68: please change "... of received signals, we can determine..." to "... of received signals, **one** can determine...". The same change should be applied in lines 69 and 71.

Line 101: "to the cosine of the zenith angle"

Line 199: "and is president in supplement...". Do the authors mean "and is presented in the supplement"?

Please make figures 5 to 8 self-contained. One should be able to understand the main message of a figure without reading the caption.

---

## Author Response (AR1)

Response to reviewer 1:

Thank you for your letter and valuable comments and questions concerning our manuscript entitled "Error analyses of a multistatic meteor radar system to obtain a 3-dimensional spatial resolution distribution" [MS no. amt-2020-353]. These comments are all valuable and very helpful for revising and improving our manuscript, as well as the important guiding significance to our researches. We have found a mistake in equation 10 and corrected it. We have studied the valuable comments from you carefully, and tried our best to revise the manuscript. These changes in the revise manuscript have been marked in the track changes version manuscript, as well as the point to point responses have listed as following:

1)The reviewer had difficulties to follow some part of the error propagation due to the various introduced coordinate systems denoted as prime without prime and so forth. This made the manuscript very hard to read and one gets easily lost.

**Response:** We apologize for our unreasonable article structure and denominations to make you confusion. Inspired by your comments, we had removed all denoted primes in coordinate system in revised manuscript. This makes our expression in manuscript more concise and readable. And the three introduced coordinate system can be well distinguished by only using subscripts.

Following your comments, we had carefully rearranged our article structure and reorganized our languages to try to make our article easy to read. We do those changes in revised version:

1. In original manuscript, we established left-hand coordinate systems with right-hand screw rule which is not idiomatical for most readers. Thus, in revised version, we change the coordinate systems to idiomatically right-hand coordinate systems with right-hand screw rule. We hope this change may increase the readability of our manuscript.

2. Section 2 is the main body of this manuscript and we divide it into four parts to make its structure more clearly. We add a brief conclusion of the analytical

process in the end of section 2 (line 275-284). And we add a flow chart to descript our analytical process (Figure 5(a)). In Figure 5(a), the variables and equations in section 2 are all included. We hope that reading section 2 while seeing Figure 5(a) will help readers understand the tedious analytical process.

3.  some units or quantitative expressions examples: use the specific angle and distance values to help readers understand the parameters settings in our program (line 282-293); use the specific location error values and resolution values to explain their relationships (line 298-301); use specific rotation angle values to explain the slant of the receiver antennas plane (line 334).

4.  Apart from correct the grammar and spelling mistakes you suggested in minor concern, we reread our manuscript to carefully check the spelling, grammar and wording. For example, "traditional meteor radars" is corrected as "classic meteor radars"; "wind retrieving" as "wind retrievals"; "AoA" and "AoAs" are unified as "AoAs"; "clockwise rotation is" as "clockwise rotation satisfies " et.al.

5.  We have found that equation 10 in original version is not correct. We have corrected it and reorganized the relative content in section 2 (line 237-268), figures and code et.al. In corrected version, except there is no "good horizontal resolution area split when baseline is long" , other results are the same.

If you have any confusion, comments or suggestions in revised manuscript, don't hesitate to feedback to us. And we would very pleasure to revise our manuscript and try to make our manuscript better. Thanks for your precious comment.

2)  Although there are some schematics outlining the coordinate systems the reviewer was not able to follow what actually is shown in Figure 5-7. The reviewer was not able to understand the plots reading the figure caption or the corresponding passage in the text. So please describe the color bars in the text or in the caption what they actually mean.

**Response:** We apologize for Figure 5-7's poor plots to make you having difficulties in reading the manuscript. Following your suggestion, we carefully replotted original manuscript's Figure 5-7 and the new figures are Figure 6-8 in revised manuscript (because we add an algorithm flow chart and is shown in Figure 5 in new manuscript , the results figures are start from Figure 6). However, due to our rearrange of original manuscript, the new figures do not correspond to original one to one. In original version, we only label the axes with coordinate axes, which is not intuitionistic. And in revised version we label the axes with noun of locality: altitude, east, north and horizontal distance. We hope this change would make readers understand the figures at a glance. In original version, there lack figure captions or corresponding text which makes the figures hard to understand. Therefore, in new version, we add more descriptions in figure captions . Because the deducing process in the section 2 is tedious, we try to provide information as much as possible in figure captions. Moreover, in Figure 6-8 we add subplots titles and colorbar unit (km) to help understand the pictures. For the reason that $E_2$ related resolution is very smaller comparing with $E_1$ related and total resolution, we change the colorbar of $E_2$ related to make this difference visible at a glance, which is not shown well in original one. Thanks very much for your comments and suggestions about our figures. If you have any other confusion, comments or suggestions about revised figures, don't hesitate to feedback to us. And we would very pleasure to carefully revise our manuscript and try to make our pictures more intuitional.

3)The reviewer understands that the authors intended to keep things as general as possible, but some units or quantitative expressions are helpful.

**Response:** Thanks very much for this very precious suggestion. We apologize for our negligence of taking some specific examples to explain some deducing processes or results that are hard to descript or understand. Using some units or

quantitative expressions are a very helpful way to increase readability. Following your suggestion, we add some units or quantitative expressions examples: use the specific angle and distance values to help readers understand the parameters settings in our program (line 282-293); use the specific location error values and resolution values to explain their relationships (line 298-301); use specific rotation angle values to explain the slant of the receiver antennas plane (line 334). If you had any other suggestions about adding some specific quantitative expressions, we would very pleasure to revise our manuscript again.

4) In particular, section 2 after line 140 is very hard to read and to follow. This is also partly the case as the Figures are only found at the end of manuscript and one has always to scroll forth and back.

**Response:** We apologize for our poor structure and presentation in section 2. Following your suggestion, we try our best to rearrange and revise section 2. Section 2 is the main body of this manuscript and we divide it into four parts to make it structure more clearly. We add a brief conclusion of the analytical process in the end of section 2 (line 275-284). We add a flow chart to descript our analytical process (Figure 5(a)). In Figure 5(a), the variables and equations in section 2 are all included. We hope that reading section 2 while seeing Figure 5(a) will help readers understand the tedious analytical process.

5) Another important point that should be discussed is that the algebraic errors are just one source that plays a role. The authors should mention in the discussion that there are other error sources as well, originating from the scattering itself or from the experimental set up due to a potential mutual antenna coupling or other obstacles in the surrounding. The later one introduces further biases in the measurements as the angle of arrivals can be significantly altered. Usually, HFSS simulation are required to investigate actually the limits of trustworthiness for the interferometry.

**Response**: Thanks very much for your suggestion. Inspired by your comments, we mention and discuss the issues of other error sources (line 348-353). The antenna design and site selection are important for meteor radars and HFSS is a powerful tool to study those issues. We only discuss the mathematic error propagation starting from phase difference measuring errors and put emphasis on multistatic configurations. We try to induce things in general, thus the discussion of some specific case of the interferometry maybe beyond the scope of our text. However, if substitute the phase difference measuring errors in our text (set as constant) to values in specific case, our method will still work(line 338-347). There are many detailed works in discuss the interferometry and their AoAs measuring errors in a more specific case, such as (Kang, 2008; Vaudrin et al.,Younger and Reid, 2017). These results of AoAs error distribution can be taken into our method to study a more specific case.

6) Furthermore, the authors should mention in the discussion that the scattering occurs not really at a singular point. The radio wave is bounced back from at least a few Fresnel zones of several kilometer length along the trajectory, which is actually most relevant for the altitude resolution as the radar signal is scattered from an extended volume (1D) and, thus, probes a volume.

**Response:** Thanks very much for your suggestions. Following your suggestions, we had carefully thought this issue. The fact that radio wave scattered from a few Fresnel zones around specular point will cause an antenna pair's phase difference deviation from the theoretical expectant value. The theoretical expectant value will resolve a AoAs pointing to specular point. This phase difference deviation is one error source of phase difference measuring errors and is included in phase difference measuring errors ($\delta(\Delta\Psi_1)$ and $\delta(\Delta\Psi_2)$). However, this issue is not clearly point out in our manuscript. Thus, we mention this issue briefly in new version (185-190, 348-350 and Figure 1-2's caption). The details of this issue can be seen in the **RC1 supplement.**

7) It is also worth to mention and discuss the issues of the sampling volume in the context of the trustworthiness of the interferometry. The schematic in Figure 2 provides a nice example of a multistatic geometry resulting in a less good measurement response compared to a monostatic radar of the same measurement volume, although the set up appears to have a multistatic geometry. The measurement response provides a measure of how well a bragg vector can be inverted to still derive reliable wind speeds (u,v,w). Ideally, all three variables can be estimated with similar measurement response, otherwise biases in one of the wind components are not avoidable. The receiver array in Figure 2 defines the sampling volume. Meteors below a certain elevation angle have to be excluded from the analysis due to the mutual antenna coupling or other ground obstacles causing issues in the interferometry.

**Response:** Very kind of you for your comments. After carefully thinking about your comments, your comments inspired us to add an important discussion about our results (354-363) to mention the issues of sampling volume and measurement response briefly. The measurement response is one of the things that affect the accuracy of Doppler shift. The location error, Doppler shift errors and other issues will determine the accuracy of the wind retrievals. We intend to discuss this in a future work. Following your suggestion, we add two black lines to represent the 30° elevation angle limit in revised figures.

8) Further, it is obvious that the angular diversity of the three links inside the remaining sampling volume is less diverse (all are located in a certain sector relative to the receiver) than a monostatic radar and could systematic bias the wind retrievals. This is the nature of the forward scatter ellipse. As all three forward scatter ellipses have the receiver site in the one of their foci points and the bragg vectors always points towards a point along the distance vector between Rx and Tx. It is further obvious that the longer the total path Rt+Rx becomes the less spatial diversity these vectors have, or with other words – all three links start to see the same geometry as it would

be the case for a monostatic radar. However, building three receiver sites and using one transmitter would increase the sampling volume and if well-distributed compensates some of this sampling effect on the wind analysis (at least partially), but still has a less good measurement response compared to a monostatic system. I suggest that they add in Figure 5-8 a line or shading area indicating the angular limit of the receiver/transmitter array by using a truncation elevation angle of about maybe 30◦. The actual limit depends on the array set up.

**Response**: Thanks very much for your comments. Following your comments, we briefly mention the issue of angular diversity (364-369). However, the impact of angular diversity of Bragg vector on wind retrievals also exceed the topic of our manuscript. We intend to discuss this in a future work. Following your suggestion, we add two black lines to represent the 30° elevation angle limit in revised figures and also mention this issue (line 319-320)

**Reference**

Ceplecha, Z., Borovička, J., Elford, W. G., ReVelle, D. O., Hawkes, R. L., Porubčan, V., and Šimek, M.: Meteor Phenomena and Bodies, Space Science Reviews, 84, 327-471, 10.1023/A:1005069928850, 1998.

Holdsworth, D. A., Reid, I. M., and Cervera, M. A.: Buckland Park all-sky interferometric meteor radar, Radio Science, 39, https://doi.org/10.1029/2003RS003014, 2004.

Hocking, W. K.: Spatial distribution of errors associated with multistatic meteor radar, Earth, Planets and Space, 70, 93, 10.1186/s40623-018-0860-2, 2018.

Kang, C.: Meteor radar signal processing and error analysis, 2008.

Vaudrin, C. V., Palo, S. E., and Chau, J. L.: Complex Plane Specular Meteor Radar Interferometry, Radio Science, 53, 112-128, 10.1002/2017rs006317, 2018.

Younger, J. P., and Reid, I. M.: Interferometer angle-of-arrival determination using precalculated phases, Radio Science, 52, 1058-1066, 10.1002/2017rs006284, 2017.

**RC1 Supplement**

**1. The issue of the radio wave scattered from Fresnel zones**

Specular meteor radars (SMR) usually utilize undersense meteor trails. (Ceplecha et al., 1998) discussed radio wave backscatter process with meteors passing though the SMR . In short, for idealized case that ignoring diffusion of meteor trail and assuming that secondary radiative and absorptive effects can be neglected, the return signal received by one antenna can be expressed as:

$$E_{R1}(x_t) = E_0 e^{i(\omega t - 2kR_0)} \int_{-\infty}^{x_t} e^{i(-\pi x^2/2)} dx \tag{1}$$

See figure 1, $R_0$ is the distance from this antenna 1 to the specular point, or the orthogonal point ($t_0$-point hereafter) in other words. $x = \sqrt{\frac{4}{\lambda R_0}} S$ and $k = \frac{2\pi}{\lambda}$. If origin time is when meteor arrives at $t_0$ point, it will get that $x_t = 2(\lambda R_0)^{-\frac{1}{2}} Vt$ (V is meteor velocity). $\int_{-\infty}^{x_t} e^{i(-\pi x^2/2)} dx$ is a complex Fresnel integral and can be expressed as $C - iS$ ,where:

$$C(x_t) = \int_{-\infty}^{x_t} \cos(\pi x^2/2) dx$$

$$S(x_t) = \int_{-\infty}^{x_t} \sin(\pi x^2/2) dx \tag{2}$$

Thus, apart from ideal specular reflection signal term "$e^{i(\omega t - 2kR_0)}$", there is a complex Fresnel modulation term $C - iS$. This modulation will cause amplitude occasion ($\sqrt{C^2 + S^2}$) and phase variation ($\phi_{add} = \arctan \frac{S}{C}$) in the period a meteor passing through. See figure 2, curve A represent the process based on eq. (1) and curve B, C, D show the effect of including an increasing degree of diffusion of the trail.

[Figure]

Figure 1

Figure 2( pick from (Ceplecha et al., 1998))

Similarly, the return signal received by antenna 2 is

$$E_{R2}(x_t) = E_0' e^{i(\omega t - kR_0 - kR_0')} \int_{-\infty}^{x_t + \Delta x_t} e^{i(-\pi x^2/2)} dx \tag{3}$$

See eq. (1) and (3), the phase difference between two antennas is from second term and third term in right side of the equations. The phase difference caused by second term is $k(R_0' - R_0)$ which is the theoretical basis of interferometer to obtain AoAs. And this phase difference will solve an AoAs pointing to specular point. However, the third term, which is related to the radio wave scattered from a few Fresnel zone, will cause additional phase difference between two antennas. This additional phase difference is caused by a delay integer length $\Delta x_t$ between two antennas. For:

$$\Delta x_t = \sqrt{\frac{4}{\lambda R_0}} \, Dsin\alpha \tag{4}$$

Take a 30MHz meteor radar for example, since $Dsin\alpha \leq 4.5\lambda$ and $R_0$ is about 100km, $\Delta x_t$ will not exceed 0.1. The major concern is how big this additional phase difference is. The change rate of the Fresnel modulation phase $\Phi$, i.e. the derivative function of $arctan(\frac{S}{C})$, will determine the magnitude of this additional phase difference. The Phase changes dramatically in pre-$t_0$ period and in small concussion after $t_0$. The additional phase difference is $\Delta x_t \frac{d\Phi}{dx_t}$ and it's no more than 25 degree around $x_t = -1$ (figure 3). Furthermore, a meteor radar system generally set an amplitude threshold to judge a meteor event and thus IQ analyze is nearly in post-$t_0$ period which additional phase is very small.

Multistatic meteor radars utilizing the forward scatter is a more general case. The effect of Fresnel zone scatter on measuring errors is nearly the same as monostatic case. See figure 4, $t_0$-point is the point where the radio wave path is shortest. Thus $t_0$-point is also the specular point where the angle of incidence equals the angle of reflection. $T_x'$ is the symmetry point of $T_x$ about meteor trail (axis-x). For a scatter point $x_i$ alongside the trail, the radio wave propagation path length is the sum of the length from $T_x'$ to $x_i$ and from $x_i$ to an antenna. Therefore $t_0$ point is the intersection of the trail path and the line from $T_x'$ to an antenna, which represents shortest path length. $t_0$ point is also specified as the origin of axis-x (or time) . For a scatter point $x_i$ which is

$S$ away from $t_0$, the radio wave propagation path length can be expressed as:

$$R = \sqrt{R_i^2 + S^2 - 2R_i S cos(90° + \theta)} + \sqrt{R_s^2 + S^2 - 2R_s S cos(90° - \theta)} \qquad (5)$$

$R_i$ and $R_s$ are specular reflection path length for incident and reflection wave. $\theta$ is the incident angle (or reflection angle). Eq. (5) can be expanded to second order because $S$ is very small compared to $R_i$ and $R_s$. Thus, R can be expressed as:

$$R = R_i + R_s + \left(cos^2 \theta \left(\frac{1}{R_i} + \frac{1}{R_s}\right)\right) S^2 \qquad (6)$$

$R_i + R_s$ correspond to $2R_0$ in monostatic case which represents the shortest path for the radio wave. If substitute $x = \sqrt{\frac{4 cos^2 \theta (R_i + R_s)}{\lambda R_i R_s}} S$, other process is the same as monostatic case.

[Figure]

Figure 3

It worth noting that a meteor trail, transmitter and receiver are not always coplanar and a meteor trail and different receiver antenna pairs are not always coplanar too. We only

give a semiquantitative analysis.

Additional phase difference and other measuring errors constitute the phase difference measuring errors ($\delta(\Delta\Psi_1)$ and $\delta(\Delta\Psi_2)$). Different radar system set different $\delta(\Delta\Psi_1)$ and $\delta(\Delta\Psi_2)$. For a receiver in Jones configuration which use at least four pairs of antennas to get AoAs, due to the phase difference measuring errors in those antennas pairs, the system should fit those four measured phase differences to get an expectant AoAs. If the RMS phase difference between the fitted and CCF phase exceeds a preselected threshold (default 20 degree) for any receiver pair the candidate is rejected (Holdsworth et al., 2004).In our program, the default value of $\delta(\Delta\Psi_1)$ and $\delta(\Delta\Psi_1)$ is 35 degree and our error propagation starts from this values. That is to say, the error that caused by the radio wave scatter from a few Fresnel zones of several kilometer length along the trajectory is included in the phase difference measuring errors ($\delta(\Delta\Psi_1)$ and $\delta(\Delta\Psi_2)$) in our analytical method .

[Figure]

Figure 4

Response to reviewer 2:

Thank your valuable comments and questions concerning our manuscript entitled "Error analyses of a multistatic meteor radar system to obtain a 3-dimensional spatial resolution distribution" [MS no. amt-2020-353]. These comments are all valuable and very helpful for revising and improving our manuscript, as well as the important guiding significance to our researches. We have found a mistake in equation 10 and corrected it. We have studied the valuable comments from you carefully, and tried our best to revise the manuscript. These changes in the revise manuscript have been marked in the track changes version manuscript, as well as the point to point responses have listed as following:

Major comments:

1)The manuscript is very difficult to follow, and the English writing needs some improvement.

**Response:** We apologize for our unreasonable article structure, English writings and denominations to make you have difficulties in reading our article. Following your comments, we carefully rearrange our manuscript and English writing trying to make our manuscript easy to understand. We do those changes in revised version:

1. we had removed all denoted primes in coordinate system in revised manuscript. This makes our expression in manuscript more concise and readable. And the three introduced coordinate systems can be well distinguished by only using subscripts. Coordinate systems or axes with prime only appear in coordinate rotations.

2. In original manuscript, we established left-hand coordinate systems with right-hand screw rule which is not idiomatical for most readers. Thus, in revised version, we change the coordinate systems to idiomatically right-hand coordinate systems with right-hand screw rule. We hope this change may increase the readability of our manuscript.

3. Section 2 is the main body of this manuscript and we divide it into four parts to make its structure more clearly. We add a brief conclusion of the analytical

process in the end of section 2 (line 275-284). And we add a flow chart to descript our analytical process (Figure 5(a)). In Figure 5(a), the variables and equations in section 2 are all included. We hope that reading section 2 while seeing Figure 5(a) will help readers understand the tedious analytical process.

4.  some units or quantitative expressions examples: use the specific angle and distance values to help readers understand the parameters settings in our program (line 282-293); use the specific location error values and resolution values to explain their relationships (line 298-301); use specific rotation angle values to explain the slant of the receiver antennas plane (line 334).

5.  Apart from correct the grammar and spelling mistakes you suggested in minor concern, we reread our manuscript to carefully check the spelling, grammar and wording. For example, "traditional meteor radars" is corrected as "classic meteor radars"; "wind retrieving" as "wind retrievals"; "AoA" and "AoAs" are unified as "AoAs"; "clockwise rotation is" as "clockwise rotation satisfies " et.al.

6.  We have found that equation 10 in original version is not correct. We have corrected it and reorganized the relative content in section 2 (line 237-268), figures and code et.al. In corrected version, except there is no "good horizontal resolution area split when baseline is long" , other results are the same.

If you have any confusion, comments or suggestions in revised manuscript, don't hesitate to feedback to us. And we would very pleasure to revise our manuscript and try to make our manuscript better. Thanks for your precious comment.

2)  Using the term "wind fields" when referring to monostatic systems is not correct. Monostatic meteor radars can be used to retrieve a mean wind vector, but not wind fields. To obtain the latter, one needs to solve for the gradients. And [du/dy], [dv/dx] can only be estimated when at least a bi-static configuration is taken into account. In connection with this issue, please re-write lines 35-50. Even with a good azimuthal sampling, the shearing term (besides the vorticity) cannot be estimated using a monostatic system. Only [du/dx], [dv/dy] can be estimated from monostatic measurements, but not [du/dy], [dv/dx]. The latter means that not only the vorticity

cannot be obtained, but neither the shearing term. Besides, there is no need to have a measure of the vertical wind in order to estimate the horizontal divergence.

**Response:** Thanks for your comments very much. We apologize for our inaccurate wording in original manuscript. Although monostatic using VAD or VVP could obtain $\frac{du}{dx}, \frac{dv}{dy}$ in certain situations, all four gradient components can not be obtained. Thus, "wind field" is not accurate for monostatic meteor radars. Following your comments, we substitute "wind field" in line 34 to "wind" for accuracy.

In original manuscript in line 35-50, we are actually discussing the case of Doppler weather radars used in troposphere measurement. We apologize for our straying from the point which had mislead you. In troposphere, atmospheric activities are strong in vertical, thus the vertical wind component projected to radial sight of the radar can not be neglected (in MLT however, the vertical wind component can be ignored). Moreover, those weather radars need to measure vertical wind components to study precipitation process of the troposphere. To obtain horizontal wind information, the vertical wind component should be removed from radial Doppler shift at first. A simple way to resolve it is using a vertical beam to detect the vertical wind. In the sampling volume, the vertical wind are assumed as the same. However, inspired by your comments, we reconsider the paragraph in line 35-50 and realize that the discussion of weather radar in this text is not suitable and will mislead the readers. Thus, we rewrite lines 35-50 only discuss the case of classic meteor radar. This makes our text more concise and keep to the point. Thanks very much for your comments.

Finally, after carefully recheck the issue of gradient components retrieving, we find that the shearing term can be obtained. Although "$\frac{du}{dy}$" and "$\frac{dv}{dx}$" can not be solved individually, their sum value (i.e. shearing term) can be obtained (Browning and Wexler, 1968). Or in other words, their subtract value (i.e. vorticity) can't be

known thus can't obtain "$\frac{du}{dy}$" and "$\frac{dv}{dx}$" individually. Details can be seen in **RC2 supplement.**

3) Instead of referring to a previous work, it would better if the authors included a simple sketch in order to understand how equation (1) is obtained.

   **Response:** Thanks for your suggestions very much. Following your suggestion, the figure 4(a) is used as a sketch to help readers understand eq.(1) and we simply explain how to obtain this equations (line 124).

4) Furthermore, the algebraic deductions of the error propagation matrixes presented in the appendixes should be treated with more care. For example, in appendix A.2, it would be helpful to have clearly indicated in its corresponding figure the angles $\gamma_1$, $\gamma_2$, $\theta$, and $\varphi$. This would help to understand, e.g., how equations A2.3 and A2.4 are obtained.

   **Response:** Very kind of you for your suggestions. We apologize for our carelessness in treating with appendix. Following your suggestions, we carefully revise the appendix.

   Some of the grammar and wordings changes are as follows:

   1. In appendix A.1 in line 461 and 463, we delete $Z'$ which will cause the misleading and substitute it with "the new coordinate".

   2. In appendix A.1 in line 471, "For any two coordinate systems $XYZ$ and $X'Y'Z'$" , we add with co-origin for more accurate.

   3. In appendix A.2 in line 479, "The AoAs is determined by two phase difference $\Delta\Psi_1$ and $\Delta\Psi_2$. Taking one antenna array as an example and Assuming " is deleted and substitute it with "In the plane wave approximation,"

   4. In original manuscript, "using Taylor expression of …" is not concise and accurate. In revised manuscript, we substitute it with "Expand $XX$X in eq.X to first order, …"

   5. In original manuscript, too many "We …" are used. In revised manuscript, we change most of them to passive voice.

Figure A.1 and A.2 are also revised following your suggestions. Figure A.1's rotation marks aren't conformed to three-dimensional perspective and will cause misleading in original version. In new version, we replot it and it can show the relationship of cover between objects. We hope this may help readers. Figure A.2 in new version adds $\theta$ and $\phi$ to help readers understand the deducing of equations.

5) In the case of appendix A.1, please modify its corresponding figure. Since the authors use lefthanded coordinate systems but follow the right-hand corkscrew rule, figure A.1 in its present form does not help to understand appendix A.1.

**Response**: In original manuscript, we established left-hand coordinate systems which is not idiomatical for most readers. Thus, in revised version, we change the coordinate systems to idiomatically right-hand coordinate systems. We hope this change may increase the readability of our manuscript. Corresponding, we modify figure A.1 in righthanded coordinate systems and follow the right-hand corkscrew rule.

6) Figures 5 to 8 contain the most important results of this work but they are poorly described and barely discussed. Besides, some of the statements based on these figures are not evident, at least for this reviewer. For example, what is stated in lines 225-226 is not obvious for the eyes of this reviewer.

Response: We apologize for Figure 5-7's poor plots to make you having difficulties in reading the manuscript. Following your suggestion, we carefully replotted original manuscript's Figure 5-7 and the new figures are Figure 6-8 in revised manuscript (because we add an algorithm flow chart and is shown in Figure 5 in new manuscript, the results figures are start from Figure 6). However, due to our rearrange of original manuscript, the new figures do not correspond to original one to one. In original version, we only label the axes with denoted coordinate axes with prime $(X_0',Y_0',Z_0')$, which is not intuitionistic. And in revised version we label the axe with noun of locality: altitude, east, north and horizontal distance. We hope this change would understand figures at a glance. In original version, there lack

figure captions or corresponding text which makes the figures hard to understand. Therefore, in new version, we add more descriptions in figure captions not only in Figure 6-8, but also Figure 1-5's Schematic diagram or flow chart. We try to provide information as much as possible in figure captions. Moreover, in Figure 6-8 we add subplots titles and colorbar unit (km) to help understand the pictures.

For the reason that $E_2$ related resolution is very smaller comparing with $E_1$ related and total resolution, we change the colorbar of $E_2$ related to make this difference visible at a glance, which is not shown well in original one. Thanks very much for your comments and suggestions about our figures. If you have any other confusion, comments or suggestions about revised figures, don't hesitate to feedback to us. And we would very pleasure to carefully revise our manuscript and try to make our pictures more intuitional.

7) This reviewer understands that the authors' objective is to analyse the errors that result from the multistatic configuration. However, the existence of other errors should be mentioned in the paper and a brief discussion on how they compare to the errors here analysed should be included. For example, it is known that the echoes do not originate on a single point in space. So, how large would it be the impact of this on the vertical resolution? Or can it be neglected?

**Response:** Thanks very much for your suggestion. Inspired by your comments, we mention and discuss the issues of other error sources (line 348-353). The antenna design and site selection are important for meteor radars and HFSS is a powerful tool to study those issues. We only discuss the mathematic error propagation starting from phase difference measuring errors and put emphasis on multistatic configurations. We try to induce things in general, thus the discussion of some specific case of the interferometry maybe beyond the scope of our text. However, if substitute the phase difference measuring errors in our text (set as constant) to values in specific case, our method will still work(line 338-347). There are many detailed works in discuss the interferometry and their AoAs measuring errors in a more specific case, such as (Kang, 2008; Vaudrin et al.,Younger and Reid, 2017). These results of AoAs error distribution can be taken into our method to study a more specific case.

Following your suggestions, we had carefully thought the issue of the radio wave scattered from Fresnel zones. The fact that radio wave scattered from a few Fresnel zones around specular point will cause an antenna pair's phase difference deviation from an ideal expectant value. The ideal expectant value will resolve a AoAs pointing to specular point. This phase difference deviation is one error source of phase difference measuring errors. Thus the impact of Fresnel scatter on measuring errors is included in phase difference measuring errors ($\delta(\Delta\Psi_1)$ and $\delta(\Delta\Psi_2)$). However, this issue is not clearly point out in our manuscript. Thus, we mention this issue briefly in new version (185-190, 348-350 and Figure 1-2's caption). The details of this issue can be seen in the **RC2 supplement.**

8) Maybe it is out of the scope of this work, but it would be helpful if some data were considered in the study. For example, what does really mean having a spatial resolution of let us say, 2-3 km? How would this impact on winds and horizontal gradients estimates? Have the authors made any rough estimation of this? It would be very useful for the readers if some information on this was included in the manuscript.

**Response**: Very kind of you for your suggestions. After carefully thinking your suggestions, your suggestions inspired us to add an important discussion about our results to briefly mention the issues of wind retrieving (line 354-369). Also, we add examples to explain the meaning of the spatial resolution, to use specific location error values and resolution values to explain their relationships (line 298-301).

The location error, Doppler shift errors and other issues will determine the accuracy of the wind retrievals. We intend to discuss this in a future work. The location error of the meteor trail's specular point, or the spatial resolution in other words, is discussed in this manuscript. Our manuscript is about 8500 word and includes the tedious analytical process with many equations. we think it would be better for our manuscript concentrate on the discussion of spatial resolutions. We will try our best to make up real data and wind retrieving discussion as soon as possible in the next.

Minor comments:

1) Line 30: please include more references here. Studies from other scientific institutions, e.g., Leipzig University and the Leibniz-IAP (Germany), which have long traditions on studies based on meteor radar measurements should be included.

   **Response:** Thanks for your suggestion. We apologize for our omissions of citing Leipzig University and the Leibniz-IAP (Germany) in line 30. Leipzig University and the Leibniz-IAP have long traditions on studying meteor radars and done many excellent works about multistatic radars in recent years. Therefore, following your suggestion we add "(Jacobi et al., 2008; Stober et al., 2013)" in line 31 in revised version.

2) Line 32: please change "… same height range be processed…" to "… same height range are processed…"

   **Response:** corrected.

3) Line 48: "Even by releasing…". I think the authors meant "relaxing".

   **Response:** corrected. Thanks for pointing out this typo.

4) Lines 53 and 59: it is MMARIA, not MMARA. Please change that.

   **Response:** corrected. Thanks for pointing out this typo.

5) Line 62: it should be "… Chau et al. used two adjacent…" and not "Stober et al."

   **Response**: corrected. Thanks for pointing out this typo.

6) Lines 65-66, what do the authors mean with "meteor radar data processing method"?

   **Response:** Thanks for your comment. Following your comment, we change "meteor radar data processing method" to "coded continuous wave meteor radar".

7) Lines 68: please change "… of received signals, we can determine…" to "… of received signals, one can determine…". The same change should be applied in lines 69 and 71.

   **Response:** corrected. Thanks for your suggestion. We also do same changes in line 151,153,155. We change the sentences using "we …" to passive voice, too.

8) Line 101: "to the cosine of the zenith angle"

   **Response**: corrected. Thanks for your suggestion.

9) Line 199: "and is president in supplement…". Do the authors mean "and is presented in the supplement"?

   **Response**: yes. Thanks for pointing out this typo. We corrected "president" to "presented".

10) Please make figures 5 to 8 self-contained. One should be able to understand the main message of a figure without reading the caption.

   **Response:** corrected. We add titles for Figures and the labels of axes are changed to "Altitude", "East", "West" and "Horizontal distance" to make figures visualized. The figure 8 in new version is a 3D contourf plot for intuitional.

$$u(x,y) = u_0 + \frac{\partial u}{\partial x}x + \frac{\partial u}{\partial y}y \tag{1}$$

$$v(x,y) = v_0 + \frac{\partial v}{\partial x}x + \frac{\partial v}{\partial y}y \tag{2}$$

$u_0$ and $v_0$ are mean wind component. Without loss of generality, the origin of coordinate-xy can be set in right above the radar. The vertical wind can be ignored.

A radial Doppler shift correspond to a radial wind velocity, denoted as $V_R(\theta, \phi)$. $\theta, \phi$ are zenith and azimuth angle of a radial direction. The unit vector in radial, denoted as $\overrightarrow{n_R}$ is:

$$\overrightarrow{n_R} = (sin\theta cos\phi, sin\theta sin\phi, cos\theta) \tag{3}$$

The wind field is projected to the radial direction and is measured by radars as $V_R$:

$$V_R(\theta, \phi) = \overrightarrow{n_R(\theta, \phi)} \cdot \overrightarrow{V(x,y)} \tag{4}$$

$$\overrightarrow{V(x,y)} = (u(x,y), v(x,y), 0) \tag{5}$$

$$(x,y) = (Htan\theta cos\phi, Htan\theta sin\phi) \tag{6}$$

simultaneous equation (1)-(6):

$$V_R(\theta, \phi) = sin\theta cos\phi u_0 + sin\theta sin\phi v_0 + Htan\theta cos\phi sin\theta cos\phi \frac{\partial u}{\partial x} +$$

$$Htan\theta sin\phi sin\theta cos\phi \frac{\partial u}{\partial y} + Htan\theta cos\phi sin\theta sin\phi \frac{\partial v}{\partial x} + Htan\theta sin\phi sin\theta sin\phi \frac{\partial v}{\partial y} \tag{7}$$

In equation (7), there are 6 variables need to be solved (mean wind and four gradient components: $u_0, v_0, \frac{\partial u}{\partial x}, \frac{\partial u}{\partial y}, \frac{\partial v}{\partial x}, \frac{\partial v}{\partial y}$). However, the coefficients ahead $\frac{\partial v}{\partial x}$ and $\frac{\partial u}{\partial y}$ are the same. This means that at most 5 variables can be obtained. By combing four and five term in right of equation (7), we can obtain:

$$V_R(\theta, \phi) = sin\theta cos\phi u_0 + sin\theta sin\phi v_0 + Htan\theta cos\phi sin\theta cos\phi \frac{\partial u}{\partial x} +$$

$$Htan\theta sin\phi sin\theta sin\phi \frac{\partial v}{\partial y} + Htan\theta sin\phi sin\theta cos\phi (\frac{\partial u}{\partial y} + \frac{\partial v}{\partial x}) \tag{8}$$

In equation (8), the five coefficient are mutually different thus five variables can be solved. They are mean wind $u_0, v_0$, two gradient components $\frac{\partial u}{\partial x}, \frac{\partial u}{\partial y}$ and shearing term $(\frac{\partial u}{\partial y} + \frac{\partial v}{\partial x})$.

**2. The issue of the radio wave scattered from Fresnel zones**

Specular meteor radars (SMR) usually utilize undersense meteor trails. (Ceplecha et al., 1998) discussed radio wave backscatter process with meteors passing though the SMR . In short, for idealized case that ignoring diffusion of meteor trail and assuming that secondary radiative and absorptive effects can be neglected, the return signal received by one antenna can be expressed as:

$$E_{R1}(x_t) = E_0 e^{i(\omega t - 2kR_0)} \int_{-\infty}^{x_t} e^{i(-\pi x^2/2)} dx \qquad (1)$$

See figure 1, $R_0$ is the distance from this antenna 1 to the specular point, or the orthogonal point ($t_0$-point hereafter) in other words. $x = \sqrt{\frac{4}{\lambda R_0}} S$ and $k = \frac{2\pi}{\lambda}$. If origin time is when meteor arrives at $t_0$ point, it will get that $x_t = 2(\lambda R_0)^{-\frac{1}{2}} Vt$ (V is meteor velocity). $\int_{-\infty}^{x_t} e^{i(-\pi x^2/2)} dx$ is a complex Fresnel integral and can be expressed as $C - iS$, where:

$$C(x_t) = \int_{-\infty}^{x_t} \cos(\pi x^2/2) dx$$

$$S(x_t) = \int_{-\infty}^{x_t} \sin(\pi x^2/2) dx \qquad (2)$$

Thus, apart from ideal specular reflection signal term "$e^{i(\omega t - 2kR_0)}$", there is a complex Fresnel modulation term $C - iS$. This modulation will cause amplitude occasion ($\sqrt{C^2 + S^2}$) and phase variation ($\phi_{add} = \arctan\frac{S}{C}$) in the period a meteor passing through. See figure 2, curve A represent the process based on eq. (1) and curve B, C, D show the effect of including an increasing degree of diffusion of the trail.

[Figure]

Figure 1

[Figure]

Figure 2( pick from (Ceplecha et al., 1998))

Similarly, the return signal received by antenna 2 is

$$E_{R2}(x_t) = E_0' e^{i(\omega t - kR_0 - kR_0')} \int_{-\infty}^{x_t + \Delta x_t} e^{i(-\pi x^2/2)} dx \qquad (3)$$

See eq. (1) and (3), the phase difference between two antennas is from second term and third term in right side of the equations. The phase difference caused by second term is $k(R_0' - R_0)$ which is the theoretical basis of interferometer to obtain AoAs. And this phase difference will solve an AoAs pointing to specular point. However, the third term, which is related to the radio wave scattered from a few Fresnel zone, will cause additional phase difference between two antennas. This additional phase difference is caused by a delay integer length $\Delta x_t$ between two antennas. For:

$$\Delta x_t = \sqrt{\frac{4}{\lambda R_0}} \; D sin\alpha \tag{4}$$

Take a 30MHz meteor radar for example, since $Dsin\alpha \leq 4.5\lambda$ and $R_0$ is about 100km, $\Delta x_t$ will not exceed 0.1. The major concern is how big this additional phase difference is. The change rate of the Fresnel modulation phase $\Phi$, i.e. the derivative function of $arctan(\frac{S}{C})$, will determine the magnitude of this additional phase difference. The Phase changes dramatically in pre-$t_0$ period and in small concussion after $t_0$. The additional phase difference is $\Delta x_t \frac{d\Phi}{dx_t}$ and it's no more than 25 degree around $x_t = -1$ (figure 3). Furthermore, a meteor radar system generally set an amplitude threshold to judge a meteor event and thus IQ analyze is nearly in post-$t_0$ period which additional phase is very small.

Multistatic meteor radars utilizing the forward scatter is a more general case. The effect of Fresnel zone scatter on measuring errors is nearly the same as monostatic case. See figure 4, $t_0$-point is the point where the radio wave path is shortest. Thus $t_0$-point is also the specular point where the angle of incidence equals the angle of reflection. $T_x'$ is the symmetry point of $T_x$ about meteor trail (axis-x). For a scatter point $x_i$ alongside the trail, the radio wave propagation path length is the sum of the length from $T_x'$ to $x_i$ and from $x_i$ to an antenna. Therefore $t_0$ point is the intersection of the trail path and the line from $T_x'$ to an antenna, which represents shortest path length. $t_0$ point is also specified as the origin of axis-x (or time) . For a scatter point $x_i$ which is $S$ away from $t_0$, the radio wave propagation path length can be expressed as:

$$R = \sqrt{R_i^2 + S^2 - 2R_i S cos(90° + \theta)} + \sqrt{R_s^2 + S^2 - 2R_s S cos(90° - \theta)} \tag{5}$$

$R_i$ and $R_s$ are specular reflection path length for incident and reflection wave. $\theta$ is the incident angle (or reflection angle). Eq. (5) can be expanded to second order because $S$ is very small compared to $R_i$ and $R_s$. Thus, R can be expressed as:

$$R = R_i + R_s + \left(cos^2 \theta \left(\frac{1}{R_i} + \frac{1}{R_s}\right)\right) S^2 \tag{6}$$

$R_i + R_s$ correspond to $2R_0$ in monostatic case which represents the shortest path for the radio wave. If substitute $x = \sqrt{\frac{4 cos^2 \theta (R_i + R_s)}{\lambda R_i R_s}} S$, other process is the same as monostatic case.

[Figure]

Figure 3

It worth noting that a meteor trail, transmitter and receiver are not always coplanar and a meteor trail and different receiver antenna pairs are not always coplanar too. We only give a semiquantitative analysis.

Additional phase difference and other measuring errors constitute the phase difference measuring errors ($\delta(\Delta\Psi_1)$ and $\delta(\Delta\Psi_2)$). Different radar system set different $\delta(\Delta\Psi_1)$ and $\delta(\Delta\Psi_2)$. For a receiver in Jones configuration which use at least four pairs of antennas to get AoAs, due to the phase difference measuring errors in those antennas pairs, the system should fit those four measured phase differences to get an expectant AoAs. If the RMS phase difference between the fitted and CCF phase exceeds a preselected threshold (default 20 degree) for any receiver pair the candidate is rejected (Holdsworth et al., 2004).In our program, the default value of $\delta(\Delta\Psi_1)$ and $\delta(\Delta\Psi_1)$ is 35 degree and our error propagation starts from this values. That is to say, the error that caused by the radio wave scatter from a few Fresnel zones of several kilometer length along the trajectory is included in the 
[revised manuscript text omitted]
^t$ points to transmitter $i$ ($T_i$). Axis-$Y_i^t$ and axis-$Z_i^t$ need to satisfy the right hand corkscrew rule with axis-$X_i^t$. Each transmitter, $T_i$, and the receiver constitute a radar link, which is referred to as $L_i$. We will deal with the range information for each $L_i$ in $X_i^t Y_i^t Z_i^t$. Spatial resolution distributions for every $L_i$ need to be compared in the same coordinate system, and this coordinate system needs to be convenient for retrieving wind fields. Therefore, we establish a local WNU (west-north-up) coordinate system $X_0^t Y_0^t Z_0^t$ on the receiver. The origin of $X_0^t Y_0^t Z_0^t$ 
[revised manuscript text omitted]

220

$$\begin{pmatrix}\delta_{(1)}X_0^t \\ \delta_{(1)}Y_0^t \\ \delta_{(1)}Z_0' \end{pmatrix} = \begin{pmatrix} X_0^t(\delta R_t) & X_0^t(\delta\theta) & X_0^t(\delta\phi) \\ Y_0^t(\delta R_t) & Y_0^t(\delta\theta) & Y_0^t(\delta\phi) \\ Z_0^t(\delta R_t) & Z_0^t(\delta\theta) & Z_0^t(\delta\phi) \end{pmatrix} \cdot \begin{pmatrix} f_R(\theta,\phi) & f_\theta(\theta,\phi) & f_\phi(\theta,\phi) \\ 0 & R_t & 0 \\ 0 & 0 & R_t\sin\theta \end{pmatrix} \cdot \begin{pmatrix} \delta R \\ \delta\theta \\ \delta\phi \end{pmatrix} \quad (10)$$

225

230  $$\begin{pmatrix} \delta R \\ \delta\theta \\ \delta\phi \end{pmatrix} = \begin{pmatrix} 1 & 0 & 0 \\ - & \frac{\frac{\lambda}{2\pi}\cos\phi}{\cos\theta\cdot D_1} & \frac{\frac{\lambda}{2\pi}\sin\phi}{\cos\theta\cdot D_2} \\ - & -\frac{\frac{\lambda}{2\pi}\sin\phi}{\sin\theta D_1} & \frac{\frac{\lambda}{2\pi}\cos\phi}{\sin\theta D_2} \\ 0 & & \end{pmatrix} \begin{pmatrix} \delta R \\ \delta(\Delta\Psi_1) \\ \delta(\Delta\Psi_2) \end{pmatrix} \quad (11)$$

235  $$\begin{pmatrix} \delta_{(1)}^2 X_0^{'} \\ \delta_{(1)}^2 Y_0^{'} \\ \delta_{(1)}^2 Z_0^{'} \end{pmatrix} = SW_{EP} \begin{pmatrix} \delta^2 R \\ \delta^2(\Delta\Psi_1) \\ \delta^2(\Delta\Psi_2) \end{pmatrix} \quad (12)$$

Here, two types of errors in different coordinate systems have been introduced. Now they need to be transformed to ENU coordinates $X_0Y_0Z_0$, which is convenient for comparing between different radar link and analysing wind fields. $E_1$ related error vectors, which are three orthogonal vectors $\overrightarrow{\delta R_s}$, $\overrightarrow{R_s\delta\theta}$ and $\overrightarrow{R_s sin\theta\delta\phi}$ and represented in XYZ as eq.(6)-(8), need to be transformed from $XYZ$ to $X_0Y_0Z_0$. To project $\overrightarrow{\delta R_s}$, $\overrightarrow{R_s\delta\theta}$ and $\overrightarrow{R_s sin\theta\delta\phi}$ towards axis-$X_0, Y_0, Z_0$ respectively and reassemble them to form three new error vectors in axis-$X_0, Y_0, Z_0$. Using coordinate rotation matrix $A_R^{(XYZ,X_0Y_0Z_0)} = A_R(\Psi_x^{i,0}, \Psi_y^{i,0}, \Psi_z^{i,0}) \cdot A_R(\psi_x^{X,i}, \psi_y^{Y,i}, \psi_z^{Z,i})$ and eq.(6)-(8), the unit vectors of those three vectors can be represented in $X_0Y_0Z_0$ as:

$$\begin{pmatrix} X_0'(\delta R_s) & X_0'(\delta\theta) & X_0'(\delta\phi) \\ Y_0'(\delta R_s) & Y_0'(\delta\theta) & Y_0'(\delta\phi) \\ Z_0'(\delta R_s) & Z_0'(\delta\theta) & Z_0'(\delta\phi) \end{pmatrix} = A_R^{(XYZ,\ X_0Y_0Z_0)} \cdot \begin{pmatrix} sin\theta cos\phi & cos\theta cos\phi & -sin\phi \\ sin\theta sin\phi & cos\theta sin\phi & cos\phi \\ cos\theta & -sin\theta & 0 \end{pmatrix} \tag{10}$$

$\left(X_0'(\delta R_s), Y_0'(\delta R_s), Z_0'(\delta R_s)\right)^T$, $\left(X_0'(\delta\theta), Y_0'(\delta\theta), Z_0'(\delta\theta)\right)^T$, $\left(X_0'(\delta\phi), Y_0'(\delta\phi), Z_0'(\delta\phi)\right)^T$ are unit vectors of $\overrightarrow{\delta R_s}$, $\overrightarrow{R_s\delta\theta}$ and $\overrightarrow{R_s sin\theta\delta\phi}$ in $X_0Y_0Z_0$ respectively.  the $3 \times 3$ matrix in left side of the eq.(10) is denoted as $P_{ij}$ for $i, j = 1,2,3$.

See eq.(6)-(8) and figure 4(c), the length of those three vectors, or error values in other words, are $\delta R_s$, $R_s\delta\theta$, $R_s sin\theta\delta\phi$ as the function of $\delta R, \delta\theta, \delta\phi$. In order to reassemble them to new error vectors, transforming $\delta\theta$ and $\delta\phi$ into two independent errors $\delta(\Delta\Psi_1)$ and $\delta(\Delta\Psi_2)$ are needed because $\delta\theta$ and $\delta\phi$ are not independent. Using eq. (3) and (4), one can transform vector $(\delta R, \delta\theta, \delta\phi)^T$ to three independent measuring errors $\delta R$, $\delta(\Delta\Psi_1)$ and $\delta(\Delta\Psi_2)$. And thus $(\delta R_s, R_s\delta\theta, R_s sin\theta\delta\phi)^T$ can be expressed as:

$$\begin{pmatrix} \delta R_s \\ R_s\delta\theta \\ R_s sin\theta\delta\phi \end{pmatrix} = \begin{pmatrix} f_R(\theta,\phi) & f_\theta(\theta,\phi) & f_\phi(\theta,\phi) \\ 0 & R_s & 0 \\ 0 & 0 & R_s sin\theta \end{pmatrix} \cdot \begin{pmatrix} 1 & 0 & 0 \\ 0 & \frac{\lambda}{2\pi} cos\phi \frac{1}{cos\theta D_1} & \frac{\lambda}{2\pi} sin\phi \frac{1}{cos\theta D_2} \\ 0 & -\frac{\lambda}{2\pi} sin\phi \frac{1}{sin\theta D_1} & \frac{\lambda}{2\pi} cos\phi \frac{1}{
[revised manuscript text omitted]

---

## Referee Report (RR1)

Second review on manuscript amt-2020-353, entitled

**Error analyses of a multistatic meteor radar system to obtain a 3-dimensional spatial resolution distribution**

By W. Zhong, X. Xue, et al.

Submitted to Atmospheric Measurements Techniques (AMT).

The authors have addressed all the points raised by this reviewer during the first review. I appreciate this very much. I think that the manuscript has improved significantly. The main results are now better discussed, and presented in clear and easy-to-read figures. Besides, other error sources are mentioned, and now it is clear that the errors due to scattering from several Fresnel zones are considered in the a priori phase difference error terms. I have only two minor concerns that should be considered before this paper can be accepted for publication.

In the first place, I am wondering if the size of the sampling grid used to do the error analysis is the proper one. 5 x 5 km in the horizontal plane seems a bit small to have good statistics in order to obtain reliable estimates of the location errors (lines 298-301 in the tracked changes file).

Second, the English writing is still not good for a prestigious journal like AMT. I mean this with all due respect, but it happened many times that I thought I was reading a telegram. There is no coherence among many sentences. Punctuation marks are missing or sometimes they are not used properly. Countless "the" articles are missing. At one point, I just stopped correcting all these mistakes, so I will only provide some examples.

The following examples apply to the line numbering that corresponds to the tracked changes file.

Line 29: change to "... detected by meteor radar**s,** regardless of **the** weather conditions".

Line 31: change to "Most modern meteor radars are monostatic**,** "

Line 39: " **viewing** direction"

Lines 53: instead of "and measurements of the non-homogenous wind fields", I suggest to write something like "and sampling the observed area from different viewing angles".

Line 59: it is better to say that multistatic meteor radars have "several" or "a few" advantages over monostatic ones, rather than "many".

Lines 67-68: what do the authors mean with "are described in the references in these papers"?

Line 69: I think the authors wanted to say "areas of interest" and not "interested areas".

Line 89: "… influence on spatial resolution distribution due to ignore the discussion of radial distance measuring error" - I think the authors mean something like "… influence on **the** spatial resolution distribution **because it** ignore**s** the  radial distance measuring error."

Line 105: "… one is those that caused by the zenith angle measuring error…" This does not make sense.

Line 106: "… and another is those that caused by the pulse length effect… " Again, this is incorrect English writing.

Line 108: "… pass**es** th**r**ough…"

Line 112: "… meteor event meanwhile…" should be changed to "… meteor event**, meaning**…"

Line 118: "in plan view"?

Line 165: "… by rotating clockwise in order of …" Did the authors mean "rotating orderly"?

Line 290: What does "… points to east by north 60°" really mean? And why 60°?

Line 300: change to "… with **equal** probability…"

As I wrote before, these are just some examples. There is an English native speaker among the authors of this manuscript. I strongly recommend that he reads the paper thoroughly and applies the needed corrections and improvements.

---

## Author Response (AR2)

Dear Editor and Reviewers:

Thank you for your letter and for reviewers' valuable comments and questions concerning our manuscript entitled "Error analyses of a multistatic meteor radar system to obtain a 3-dimensional spatial resolution distribution" [MS no. amt-2020-353]. We have studied the valuable comments from you and reviewers carefully, and tried our best to revise the manuscript. I sincerely apologize for my poor English writing that have made revised manuscript hard to read. Therefore, Prof. Reid and other coauthor had polished manuscript again. We hope that you could be satisfied with the new revision. These changes in the revise manuscript have been marked blue in the track changes version manuscript, as well as the point to point responses have listed as following:

**Response to reviewer #1:**

General comments:

1) Only the issue of mutual antenna coupling was not picked up in detail.

**Response:** We are sorry for our insufficient discussion about mutual antenna coupling. We only limit the elevation angle to avoid the effect of mutual antenna coupling, although HFSS can evaluate this effect. We mainly discuss the error propagation in general situation and due to space limitations, we only simply mention this effect. However, in the future work that discuss a specific radar system, we will discuss some issues suggested by you including mutual antenna coupling in detail. For example, we intend to do some work about meteor radar detection in low altitude and low elevation angle, thus HFSS is a powerful tool for our study. Thanks for your suggestion about HFSS very much.

Minor suggestions/technical corrections:

1) Vertical wind (line 33-34):

This statement is not true. There are publications that do not use this assumption, but there is no validation available whether these winds are scientifically useful (see Egito et al., 2016, Stober et al., 2020, Stober et al., 2021, Conte et al., 2021, Chau et al., 2020, etc). Considering the huge systematic issues in stated values, it is recommended to weaken a bit the statement. But vertical winds provide a very good quality assessment of the retrieval and, thus are important.

**Response:** Very kind of you for your comments and suggestion. Following your suggestion, we add "usually" in front of "a zero wind" to weaken a bit statement.

2) MIMO, SIMO, etc (line 53-55):

The first statistical MIMO radar was not Chau et al., 2019 a better citation is the French RIAS system. It operated in the meteor wavelength range and consisted of two circular arrays. However, the focus was not directly on atmospheric dynamics, but the concept is rather old. Please add the citation. J. Dorey, Y. Blanchard, F. Christophe, "Le projet RIAS, une approche nouvelle du radar de surveillance aérienne", Colloque International dur le radar, 1984, Versailles, France.

**Response:** Thanks for your comments very much. We add this reference in new version.

3) Discussion of measurement response (line 295-310):

The reviewer provided a statement in his review about the measurement response. The reply of the authors and revising in the manuscript is appreciated. The reviewer apologizes for being not specific enough on that topic. The paragraph requires no changes, although the term measurement response is not used in its mathematical contest. The measurement response describes how much an observation reduces an a priori covariance for a certain state. The measurement response provides a mathematical assessment of the observations, the forward model, and all error contributions intrinsic to the observations. However, this goes beyond the paper.

**Response:** Thanks for your comments very much. The issues about measurement response and the priori covariance et.al. will be emphatically discussed in our future work. These issues are important for wind retrieving. In line 353-365 in tracked changes version, we reorganized the language to make it more accurate.

4) Conclusion (line 324) :

The statement about SIMO is questionable as the paper does not investigate any advantages or disadvantages between the systems. So please remove the statement advantage. Time will tell which system advantageous.

**Response:** Thanks for your comments very much. We apologize for our unreasonable statement. Following your suggestion, we change the sentence in line 381-382 to "also show good experimental results "

**Response to reviewer #2:**

1) In the first place, I am wondering if the size of the sampling grid used to do the error analysis is the proper one. 5 x 5 km in the horizontal plane seems a bit small to have good statistics in order to obtain reliable estimates of the location errors (lines 298-301 in the tracked changes file).

**Response:** Thanks for your comments very much. The selection of a suitable "horizontal sampling measurement area" for multistatic meteor radars is a good question. As Figure 8 shows, a meteor event's horizontal location error can exceed

15km or more in some effective measuring region. Therefore, take (Stober, et.al. 2018) for example, the sampling area in horizontal is 30 x 30km. However, the "horizontal sampling grid length" in our manuscript is a default calculation length in program. This sampling grid length can be set to lager value to reduce computational cost or smaller to make contourf more elaborate. But the calculation results, that's to say, spatial resolutions, will determine the values of suitable "horizontal sampling measurement area". In the region where horizontal resolution is low, "horizontal sampling measurement area" will chose to be larger. While in high resolution region, "horizontal sampling measurement area" could chose to be smaller. The issues of selecting a suitable "horizontal sampling measurement area" will be discussed in future work.

2) Second, the English writing is still not good for a prestigious journal like AMT. I mean this with all due respect, but it happened many times that I thought I was reading a telegram. There is no coherence among many sentences. Punctuation marks are missing or sometimes they are not used properly. Countless "the" articles are missing. At one point, I just stopped correcting all these mistakes, so I will only provide some examples.

**Response:** Thanks for your suggestions very much. The revised manuscript is changed a lot comparing with original one and didn't polished well for time's sake. Thereby, I sincerely apologize for my poor English writing that have made revised manuscript hard to read. Prof. Reid and other coauthor had helped me polished manuscript again. We hope that you could be satisfied with the new revision.

The following examples apply to the line numbering that corresponds to the tracked changes file.

1) Line 29: change to "... detected by meteor radars, regardless of the weather conditions".

corrected

2) Line 31: change to "Most modern meteor radars are monostatic."

corrected

3) Line 39: "look viewing direction"

corrected

4) Lines 53: instead of "and measurements of the non-homogenous wind fields", I suggest to write something like "and sampling the observed area from different viewing angles".

Corrected. Thanks for your suggestion.

5) Line 59: it is better to say that multistatic meteor radars have "several" or "a few" advantages over monostatic ones, rather than "many".

Corrected. Thanks for your suggestion.

6)  Lines 67-68: what do the authors mean with "are described in the references in these papers"?

We apologize for our language errors that mislead you. In line 67-68 in tracked changes version, we correct it to "are described in the corresponding references."

7) Line 69: I think the authors wanted to say "areas of interest" and not "interested areas".

Yes. Corrected.

8) Line 89: "… influence on spatial resolution distribution due to ignore the discussion of radial distance measuring error" - I think the authors mean something like "… influence on the spatial resolution distribution because it ignores the discussion of radial distance measuring error."

Yes. Corrected.

9) Line 105: "… one is those that caused by the zenith angle measuring error…" This does not make sense. Line 106: "… and another is those that caused by the pulse length effect… " Again, this is incorrect English writing.

Thanks for you comments. In new version, we corrected it to "one caused by…, and one caused…"

10) Line 108: "… passes through…"

Corrected.

11) Line 112: "… meteor event meanwhile…" should be changed to "… meteor event, meaning…"

Corrected.

12) Line 118: "in plan view"?

Means "", maybe better

13) Line 165: "… by rotating clockwise in order of …" Did the authors mean "rotating orderly"?

Yes, Corrected.

14) Line 290: What does "… points to east by north 60°" really mean? And why 60°?

Thanks for you comments. In manuscript we take "$\psi_x^{i,0} = \psi_y^{i,0} = 0$, $\psi_z^{i,0} = 30°$ and $d_i = 250km$; $\psi_x^{X,i} = 5°, \psi_y^{Y,i} = 0, \psi_z^{Z,i} = 0$" for example. "$\psi_x^{i,0} = \psi_y^{i,0} = 0$, $\psi_z^{i,0} = 30°$" means that axis-$X_i$ points to $30°$ east by south thus axis-$Y_i$ points to east by north $60°$. "$\psi_x^{X,i} = 5°, \psi_y^{Y,i} = 0, \psi_z^{Z,i} = 0$" means that axis-X is collinear with axis-$X_i$ and axis-Y has $5°$ elevation angle right above axis-$Y_i$. Se image below.

[Figure]

15) Line 300: change to "… with equal probability…"

Corrected.

[revised manuscript text omitted]